# What Went Wrong? Closing the Sim-to-Real Gap via Differentiable Causal Discovery

**Peide Huang**[1], **Xilun Zhang**[1*], **Ziang Cao**[1*], **Shiqi Liu**[1*]

**Mengdi Xu**[1], **Wenhao Ding**[1], **Jonathan Francis**[2], **Bingqing Chen**[2], **Ding Zhao**[1]

[1]Carnegie Mellon University, [2]Bosch Center for Artificial Intelligence, *equal contribution
{peideh, xilunz, ziangc, shiqiliu, mengdixu, wenhaod, dingzhao}@andrew.cmu.edu
{jon.francis, bingqing.chen}@us.bosch.com

**Abstract:** Training control policies in simulation is more appealing than on real robots directly, as it allows for exploring diverse states in an efficient manner. Yet, robot simulators inevitably exhibit disparities from the real-world dynamics, yielding inaccuracies that manifest as the dynamical simulation-to-reality (sim-to-real) gap. Existing literature has proposed to close this gap by actively modifying specific simulator parameters to align the simulated data with real-world observations. However, the set of tunable parameters is usually manually selected to reduce the search space in a case-by-case manner, which is hard to scale up for complex systems and requires extensive domain knowledge. To address the scalability issue and automate the parameter-tuning process, we introduce COMPASS, which aligns the simulator with the real world by discovering the causal relationship between the environment parameters and the sim-to-real gap. Concretely, our method learns a differentiable mapping from the environment parameters to the differences between simulated and real-world robot-object trajectories. This mapping is governed by a simultaneously learned causal graph to help prune the search space of parameters, provide better interpretability, and improve generalization on unseen parameters. We perform experiments to achieve both sim-to-sim and sim-to-real transfer, and show that our method has significant improvements in trajectory alignment and task success rate over strong baselines in several challenging manipulation tasks. Demos are available on our project website: https://sites.google.com/view/sim2real-compass.

**Keywords:** sim-to-real gap, reinforcement learning, causal discovery

## 1  Introduction

Training control policies directly on real robots poses challenges due to the sample complexity of deep reinforcement learning (RL) algorithms. Therefore, training in simulation is often necessary to perform diverse exploration of the state-action space in an efficient manner [1, 2, 3, 4, 5, 6, 7]. However, robot simulators are constructed based on simplified models and are thus approximations of the real world. For example, dynamics such as contact and collision are notoriously difficult to simulate with simplified physics [8, 9]. Even if the dynamics could be simulated accurately, not all physical parameters can be precisely measured in the real world and specified in simulation, e.g., friction coefficients, actuation delay, etc. As a result, a robot that is trained in a biased simulator could have catastrophic performance degradation in the real world [10, 11, 12, 13]. It is, therefore, critical to use simulators that closely mimic real-world dynamics to reduce this sim-to-real gap [14, 15, 16, 17].

Existing literature has proposed to close the sim-to-real gap by adjusting the parameters of the simulator to align the simulated data with the observed real data. To facilitate this, robot simulators such

7th Conference on Robot Learning (CoRL 2023), Atlanta, USA.

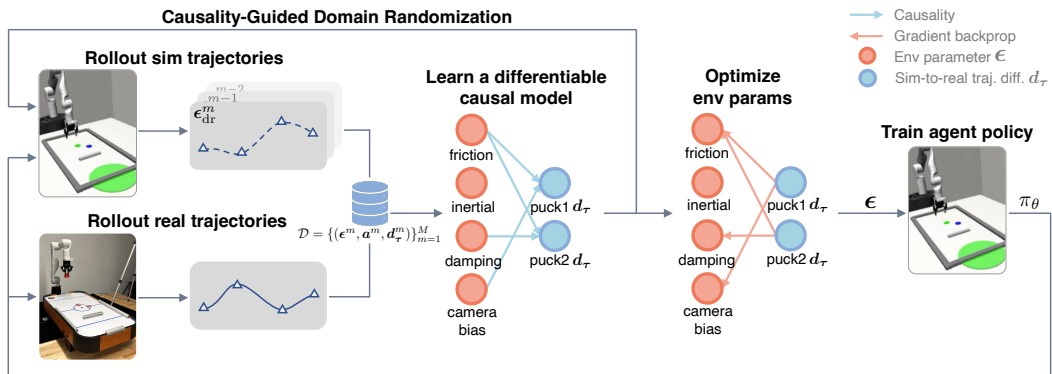

Figure 1: Overview of the COMPASS framework.

as robosuite [18] provide APIs to modify over 400 different environment parameters. Unfortunately, the search space grows exponentially with the dimension of the environment parameters. To mitigate this issue, various existing methods attempt to modify the parameters more efficiently, using gradient-based or gradient-free sampling-based techniques [19, 20]. For example, in quasi-static manipulation tasks, such as sorting pegs or opening drawers, Chebotar et al. [19] chose to modify parameters related to the size, positions, and compliance; in dynamic manipulation tasks, Muratore et al. [20] chose to modify mass, friction, and restitution coefficients, etc.

However, this parameter selection process is typically carried out on a case-by-case basis, necessitating substantial domain knowledge to restrict the scope of environment parameters. This becomes challenging when dealing with simulations involving multiple interacting objects [21]. Furthermore, most existing methods lack the capability to offer explicit insights that can effectively guide the future deployment of more complex systems. They fail to offer direct answers to the question "*What went wrong with my simulator*" or, more specifically, "*What simulator parameters should I tune to reduce the sim-to-real gap*" without *post hoc* analysis of the final parameters. In contrast, humans are good at analyzing complex events, identifying and eliminating irrelevant factors, and uncovering crucial cause-and-effect relationships. Such causal discovery capability enables an efficient and interpretable search [22, 23, 24] for differences between two systems, providing a promising direction to bridge the gap between simulation and reality.

In this work, we propose a method that aims to align the simulator with the real world by discovering the **c**ausality between envir**onm**ent **pa**rameter**s** and the **s**im-to-real gap (COMPASS) as illustrated in Figure 1. COMPASS learns a differentiable mapping, from the simulation environment parameters to the differences between simulated and real-world trajectories of dynamic robot-object interactions, governed by a simultaneously-learned causal graph. With the differentiable causal model fixed, COMPASS back-propagates gradients to optimize the simulation environment parameters in an end-to-end manner to reduce the domain gaps. Beyond the interpretability, the causal graph also helps to prune the parameter search space, thus improving the efficiency of domain randomization as well as the scalability. We summarize our contributions as follows:

1. We propose a novel causality-guided parameter estimation framework to close the sim-to-real gap and improve agent performance in the real world.

2. We design a fully-differentiable model that explicitly embeds the causal structure to provide better interpretability, prune the search space of parameters, and improve generalization.

3. We empirically evaluate our method in both the simulation and the real world, which outperforms baselines in terms of trajectory alignment and task success rate with the same sample size.

## 2 Related Work

Closing the sim-to-real gap in robotic control tasks is often approached through domain randomization (DR) [10, 25, 26, 27, 28, 29, 30, 31]. Although DR proves successful in numerous applications, particularly when access to real environments or data collected therein is unavailable, it is recognized

that vanilla DR may result in overly conservative policies when the range of randomization is broad [32, 33]. As an alternative approach for achieving sim-to-real transfer, system identification aims to estimate the parameters of the environment through limited interactions with real environments [20, 34, 21, 35, 36, 37] and has been combined with DR methods. We draw inspiration from this line of work in developing our parameter estimation framework, which learns causal relationships from dynamic robot-object interactions in order to facilitate the sim-to-real transfer.

**Gradient-free Parameter Estimation for Sim-to-Real Transfer.** Gradient-free parameter estimation methods typically utilize sampling-based methods to update the simulation environment parameters. Chebotar et al. [19] propose the SimOpt framework, which iteratively alters the distribution of environment parameters in simulation to mirror real environment rollouts via the cross-entropy method. Moving away from the assumption that the distribution of environment parameters follows a Gaussian distribution, as adopted in [19], Ramos et al. [38] develop BayesSim, which uses a Gaussian mixture model and optimizes the parameter distribution from a Bayesian perspective. Muratore et al. [20] propose Neural Posterior Domain Randomization (NPDR) which further removes assumptions on the environment parameter distribution by utilizing neural likelihood-free inference methods and could handle correlated parameters. It is worth noting that scaling up sampling-based methods becomes challenging when dealing with a large number of parameters. In contrast, `COMPASS` learns a difference-prediction model, leverages gradients to adjust the simulation parameters, and further improves scalability with learned causal structures.

**Gradient-based Parameter Estimation for Sim-to-Real Transfer.** Gradient-based methods typically employ a neural network to encapsulate the gradient landscape of parameter differences [34, 21, 35] or to model environment dynamics [36]. TuneNet [34] uses a neural network to predict the discrepancies in parameters based on the observations derived from two distinct environments. The Search Parameter Model (SPM) [35] is a binary classifier, with rollouts and parameters as input, which predicts whether a set of parameters is higher or lower than the target ones. Unlike TuneNet or SPM, we opt to predict observation differences using environment parameters as inputs. Allevato et al. [21] further expand the capabilities of TuneNet from handling a single parameter to managing a model with 5 parameters. In contrast, we demonstrate `COMPASS` is able to scale up to a 64-parameter system, a capacity notably larger than existing works. EXI-Net [36] implements a dynamics predictive model, conditioned on environment parameters, and identifies the most suitable parameters via back-propagation. While EXI-Net strives to model a broad spectrum of environments by separately modeling the known/explicit and implicit dynamics parameters, we aim to enhance sim-to-real transfer efficiency by capitalizing on the progressively discovered causal structure.

## 3 Methodology

### 3.1 Problem formulation: Markov Decision Processes and sim-to-real gap

A finite-horizon Markov Decision Process (MDP) is defined by $\mathcal{M} = (\mathcal{S}, \mathcal{A}, P, R, p_0, \gamma, T)$, where $\mathcal{S}$ and $\mathcal{A}$ are state and action spaces, $P : \mathcal{S} \times \mathcal{A} \times \mathcal{S} \rightarrow \mathbb{R}_+$ is a state-transition probability function or probabilistic system dynamics, $R : \mathcal{S} \times \mathcal{A} \rightarrow \mathbb{R}$ is a reward function, $p_0 : \mathcal{S} \rightarrow \mathbb{R}_+$ is an initial state distribution, $\gamma$ is a reward discount factor, and $T$ is a fixed horizon. Let $\tau = (s_0, a_0, \ldots, s_T, a_T)$ be a trajectory of states and actions and $R(\tau) = \sum_{t=0}^{T} \gamma^t R(s_t, a_t)$ is the trajectory reward. The objective of RL is to find parameters $\theta$ of a policy $\pi_\theta(a|s)$ that maximize the expected discounted reward over trajectories induced by the policy: $\mathbb{E}_{\pi_\theta}[R(\tau)]$, where $s_0 \sim p_0, s_{t+1} \sim P(s_{t+1}|s_t, a_t)$, and $a_t \sim \pi_\theta(a_t|s_t)$.

In our work, we assume that the simulator's system dynamics are conditioned on environment parameters $\boldsymbol{\epsilon} \in \mathbb{R}^{|\mathcal{E}|}$, i.e., $P : \mathcal{S} \times \mathcal{A} \times \mathcal{S} \times \mathbb{R}^{|\mathcal{E}|} \rightarrow \mathbb{R}_+$, where $\mathcal{E}$ is the set of all tunable environment parameters and $|\cdot|$ measures the cardinality of the set. Given a simulator parameterized by $\boldsymbol{\epsilon}$, the agent is optimizing $\mathbb{E}_{\pi_\theta, \boldsymbol{\epsilon}}[R(\tau)]$. When the simulation dynamics are very close to the real-world dynamics, one can expect the trajectory rollouts in the simulator to be close to that in the real world as well. Hence, an optimal agent trained in the simulator would expect near-optimal performance in the real world [39]. However, due to unmodeled dynamics and inaccurate environment parameters, the

simulation dynamics are different from the real world (i.e., there exists a sim-to-real gap), resulting in different trajectory rollouts and thus degradation in real-world performance [1, 2, 10, 11, 15].

For better interpretability, we assume a factorized state space, i.e., $\mathcal{S} = \{\mathcal{S}_1 \times \cdots \times \mathcal{S}_K\}$, with $s_{k,t} \in \mathcal{S}_k$ representing the $k$-th factorized state at time $t$. Each component usually has explicit semantic meanings (i.e., an event or object's property) [23], which holds through state and action abstraction in general [40, 41, 42]. For example, in the case of *pick-and-place*, the state space can be factorized to the 3D position and orientation of the object and end effector. Similar to Chebotar et al. [19], we then define a factorized trajectory difference function:

$$d_k(\tau_{\text{sim}}, \tau_{\text{real}}) := \sum_{t=0}^{T} \|s_{k,t,\text{sim}} - s_{k,t,\text{real}}\|_2, \quad \text{for } k = 1, 2, \ldots, K \tag{1}$$

The trajectory difference function is then $\boldsymbol{d} := [d_1, \ldots, d_K]$, and the trajectory difference $\boldsymbol{d_\tau}$ is the output of $\boldsymbol{d}$ to measure the sim-to-real gap between of a pair of trajectories, $(\tau_{\text{sim}}, \tau_{\text{real}})$. In this work, we aim to find a simulation environment parameter $\epsilon$ that minimizes the expectation of trajectory differences $\boldsymbol{d_\tau}$ under the same policy.

## 3.2 Learning causality between environment parameters and trajectory differences

To model the causality, COMPASS learns a causal model $f_\phi(\epsilon, \boldsymbol{a}; \mathcal{G})$ mapping the environment parameter $\epsilon$ and action sequence $\boldsymbol{a} = [a_0, \ldots, a_T]$ to the trajectory difference $\boldsymbol{d_\tau}$. This model contains a causal graph $\mathcal{G}$, whose nodes represent the variables to be considered and the edges represent the causal influence from one node to another node. We jointly learn the model parameter $\phi$ and discover the underlying causal graph $\mathcal{G}$ in a fully differentiable manner.

**Causal Graph.** The causal graph $\mathcal{G}$ plays a crucial role in the model by providing interpretability, pruning the search space of parameters, and improving the generalization on unseen parameters. Since we focus on the influence of environment parameter $\epsilon$ to the trajectory difference $\boldsymbol{d_\tau}$, we can represent the graph with a binary adjacency matrix of size $|\mathcal{E}| \times K$, where $1/0$ indicates the existence/absence of an edge from the environment parameter to the trajectory difference. Motivated by previous works [43, 44, 45] that formulate the combinatorial graph learning into a continuous optimization problem, we design a sample-efficient pipeline by making the optimization of $\mathcal{G}$ differentiable. We sample elements of the graph $\mathcal{G}$ from a Gumbel-Softmax distribution [46], parametrized by $\psi \in [0, 1]^{|\mathcal{E}| \times K}$, i.e., $\mathcal{G}_{ij} \sim \texttt{GumbelSoftmax}(\psi_{ij}; \mathcal{T} = 1)$, where $\mathcal{T}$ is the softmax temperature. We denote the parameterized causal graph as $\mathcal{G}_\psi$. All elements $(i, j)$ are initialized to ones to ensure the causal graph is fully connected at the beginning.

**Structural Causal Model.** Since the causal graph only describes the connection between variables, we also need a parameterized model to precisely represent *how* the causes influence the effects. We design an encoder-decoder structure in $f$, with $\mathcal{G}$ as a linear transformation applied to the intermediate features. First, the encoder operates on each dimension of $\epsilon$ independently to generate features $z_\epsilon \in \mathbb{R}^{|\mathcal{E}| \times d_z}$. Then the causal graph is multiplied by the features to generate the inputs for the decoder, i.e., $g_\epsilon = z_\epsilon^T \mathcal{G} \in \mathbb{R}^{d_z \times K}$, where $d_z$ is the dimension of the feature. Similarly, the action sequence $\boldsymbol{a}$ is passed through the encoder and transformation to produce the feature of the action sequence $g_{\boldsymbol{a}} \in \mathbb{R}^{d_z \times K}$. Finally, $g_\epsilon + g_{\boldsymbol{a}}$ is passed through the decoder to output the prediction $\hat{\boldsymbol{d}_\tau}$.

**Differentiable Causal Discovery.** Given a dataset $\mathcal{D} := \{\epsilon^m, \boldsymbol{a}^m, \boldsymbol{d}_\tau^m\}_{m=1,\ldots,M}$, the optimization objective to discover the underlying causal model consists of two terms:

$$\mathcal{L}_{\phi,\psi} := \frac{1}{M} \sum_{m=1}^{M} \|f_\phi(\epsilon^m, \boldsymbol{a}^m; \mathcal{G}_\psi) - \boldsymbol{d}_\tau^m\|_2^2 + \lambda \|\psi\|_p^p, \tag{2}$$

where the first term is the mean squared error between the predicted trajectory differences and the real differences, and the second is a regularization term that encourages the sparsity of $\mathcal{G}$ ($\|\psi\|_p$ is the entry-wise p-norm of $\psi$) with a positive scalar $\lambda$ to eliminate the influence of irrelevant environment parameters. The detailed architecture of this causal model can be found in Appendix A.

**Algorithm 1** Causality between envirOnMent PArameterS and the Sim-to-real gap (COMPASS)

---

1: **Input:**
2: $\epsilon_0 \in \mathbb{R}^{|\mathcal{E}|}$: initial guess of environment parameters,
3: $\zeta$: threshold for sim-to-real gap,
4: $\text{Sim}(\cdot)$: simulator with controllable environment parameters
5: **Output:** $\epsilon_i, \pi_\theta$

---

6: Initialize agent policy $\pi_\theta$
7: Initialize $f_\phi(\epsilon, a; \mathcal{G}_\psi)$ with $\psi \leftarrow \mathbb{1}_{|\mathcal{E}| \times K}$
8: **for** $i = 0, 1, 2, \ldots, MaxIter$ **do**
9:      Train $\pi_\theta$ in $\text{Sim}(\epsilon_i)$
10:      $\{\tau_{\text{sim}}^n\}_{n=1,\ldots,N} \leftarrow$ Rollout $N$ trajectories using $\pi_\theta$ in $\text{Sim}(\epsilon_i)$
11:      $\{\tau_{\text{real}}^n\}_{n=1,\ldots,N} \leftarrow$ Rollout $N$ trajectories using $\pi_\theta$ in the real environment
12:      Stop the iterations **if** $\text{AVERAGE}(d(\tau_{\text{sim}}^1, \tau_{\text{real}}^1), \ldots, d(\tau_{\text{sim}}^N, \tau_{\text{real}}^N)) \leq \zeta$
13:      $\mathcal{D} \leftarrow \varnothing$
14:      **for** $n \in \{1, \ldots, N\}$ **do**                           ▷ This loop can run in parallel
15:          $\{\epsilon_{\text{dr}}^m\}_{m=1,\ldots,M} \leftarrow$ CAUSALITYGUIDEDDOMAINRANDOMIZATION$(\epsilon_i, \psi)$
16:          **for** $m \in \{1, \ldots, M\}$ **do**
17:              $\tau_{\text{sim}}^m \leftarrow$ Rollout $\pi_\theta$ in $\text{Sim}(\epsilon_{\text{dr}}^m)$
18:              $d_\tau^m \leftarrow d(\tau_{\text{sim}}^m, \tau_{\text{real}}^n)$
19:              $\mathcal{D} \leftarrow \mathcal{D} \cup \{\epsilon_{\text{dr}}^m, \tau_{\text{real}}^n, d_\tau^m\}$
20:      Jointly optimize model parameter $\phi$ and causal graph parameter $\psi$ of $f_\phi(\epsilon, a; \mathcal{G}_\psi)$    ▷ Eq. 2
21:      $\epsilon_{i+1} \leftarrow$ UPDATEENVPARAM$(\epsilon_i, f_\phi(\epsilon_i, a; \mathcal{G}_\psi))$                        ▷ Eq. 3

---

**Algorithm 2** Causality-Guided Domain Randomization

---

1: **function** CAUSALITYGUIDEDDOMAINRANDOMIZATION$(\epsilon, \psi)$
2:      **for** $r = 1, 2, \ldots, |\mathcal{E}|$ **do**
3:          **if** $\max(\psi_r) > Threshold$ **then**                 ▷ $\psi_r$ is the $r$-th row of $\psi$
4:              $\{\epsilon_r^m\}_{m=1,\ldots,M} \leftarrow$ UNIFORM$(\epsilon_r - \delta_r, \epsilon_r + \delta_r)$     ▷ $\epsilon_r$ is the $r$-th dimension of $\epsilon$
5:      **return** $\{\epsilon^m\}_{m=1,\ldots,M}$

---

### 3.3 Closing the sim-to-real gap via differentiable causal discovery

The main algorithm is shown in Algorithm 1. We highlight two parts of the algorithm here.

**Causality-guided Domain Randomization.** The learned causal graph is used to prune the search space. Since each element of the causal graph parameter $\psi$ indicates the probability of an edge from the environment parameter to the trajectory difference, we can use $\psi$ to determine whether to randomize a particular environment parameter or not. For instance, if the learned $\psi$ indicates that there is no causal relationship between the torsional friction and the trajectory difference, the torsional friction will be excluded from randomization in the subsequent iteration, which enhances the efficiency of DR. The randomized environment parameters are sampled uniformly from certain ranges according to the current environment parameters. In this way, COMPASS automatically reduces the search space by orders of magnitude without human supervision or domain knowledge. The detail of causality-guided domain randomization is shown in Algorithm 2.

**Environment Parameter Optimization.** Owing to the full differentiability of our model, we can back-propagate the gradient information directly to the environment parameters to minimize the predicted sim-to-real gap:

$$J = \frac{1}{K} \sum_{k=1}^{K} f_{\phi,k}(\epsilon, a; \mathcal{G}_\psi), \quad \epsilon \leftarrow \epsilon - \eta \nabla_\epsilon J \tag{3}$$

where $f_{\phi,k}$ is the $k$-th dimension of the output. We use the real action sequences and apply Eq. 3 multiple times until convergence or reaching the maximum step. The sparse causal graph $\mathcal{G}$ could improve the robustness against noisy trajectory data and generalization on unseen environment parameter values during parameter optimization.

# 4 Experimental Results

## 4.1 Experimental setups

**Mini-Air-Hockey with Obstacle.** We design a challenging task of playing air hockey with a robot arm. To reach the goal, the agent needs to consider pusher-to-puck, puck-to-puck, puck-to-wall collisions, and surface properties of the hockey table. The task is to manipulate the pusher to hit the first puck, colliding with the second, which in turn needs to avoid the obstacle to reach the goal position by bouncing against the wall. Similar to Evans et al. [47], the action space includes the starting position of the pusher, hitting angle, and velocity. The state space includes the position of the two pucks ($K = 2$). The pusher, puck, and goal have a radius of 3cm, 2.55cm, and 15cm, respectively. This task requires precise actuation since objects interact multiple times, propagating and compounding sim-to-real mismatches such that the agent can experience a significant

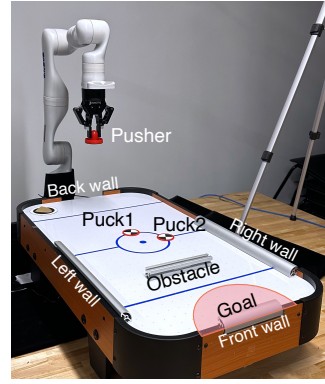

Figure 2: Experimental setup.

drop in success rate. Additionally, methods need to be robust against noisy and unmodeled dynamics. For instance, the floating force, in reality, is generated by a fan placed at the center of the table, and thus is non-uniform and stochastic. There are 64 tunable environment parameters in our experiments ($|\mathcal{E}| = 64$), and we use parameter notations in the format of *object@param_type@param*. We use the Kinova Gen 3 robot arm in the real-world test bench and simulate in the `robosuite` [18] environment with MuJoCo physics engine [8]. Experimental details and two supplementary experiments, **Double-Bouncing-Ball** and **Push-I**, are available in Appendix B, E and F, respectively.

**Baselines.** As baselines, we select the state-of-the-art gradient-free sampling baselines NPDR [20], and two gradient-based baselines, TuneNet [34] and EXI-Net [36], as discussed in Section 2.

## 4.2 Sim-to-sim trajectory alignment with known target environment parameters

In this experiment, we verify whether `COMPASS` can align trajectories between two different environments. We conduct experiments in simulation so that the ground truth environment parameters are known to us. Among the two environments, one is treated as the "real" (target) environment we want to align with, the other is the simulation environment we will fine-tune. We collect rollouts with a scripted stochastic policy. For each method except Tune-Net, we use *MaxIter* $= 10, N = 10, M = 64$. For Tune-Net, we collect a dataset of size *MaxIter* $\times N \times M = 6400$ to train the regression model. More implement details and ablation study are presented in Appendix C and D.

The learned causal graph parameters $\psi$ after 2 iterations ($i = 0, 1, 2$) are shown in Figure. 3. We observe that the learned causal graph is very sparse, reducing the search space by orders of magnitude without extensive domain knowledge. Our method is able to automatically discover different types of relevant environment parameters such as actuation, sensing, and dynamics. It is worth noting that only the damping of the right wall out of 4 sides has causality discovered with the trajectory

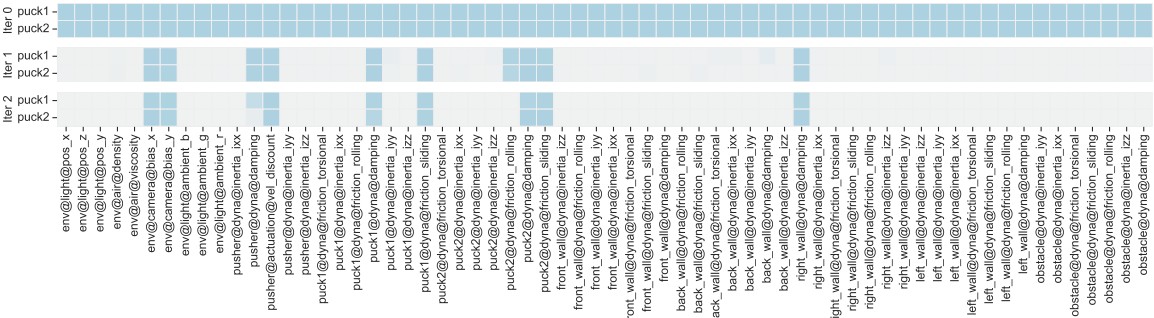

Figure 3: Learned causal graph parameters $\psi$. Darker colors present values closer to 1. We use parameter notations in the format of *object@param_type@param*.

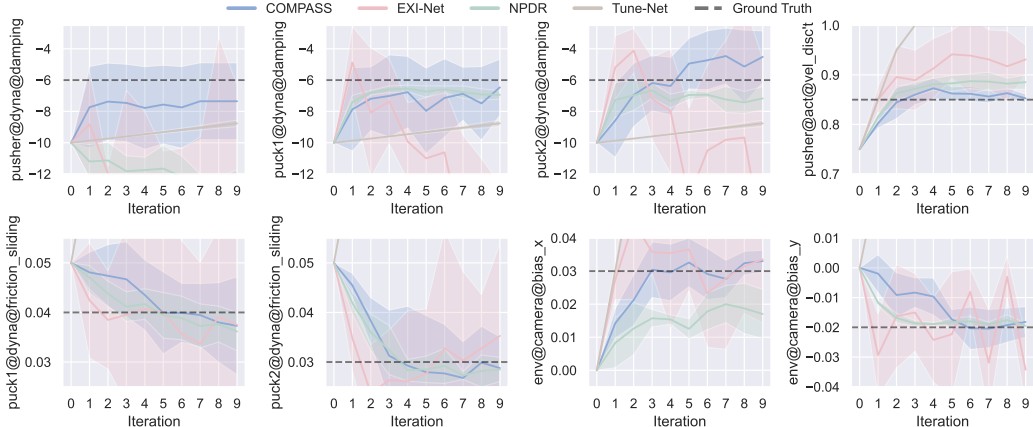

Figure 4: Environment parameters optimization. We show 8 parameters with discovered causality (as shown in Figure 3). The final mean absolute percentage errors (MAPE) are 0.22, 0.95, 0.29, 3.85 for COMPASS, EXI-Net, NPDR, Tune-Net, respectively. The solid lines represent the mean value across 5 random seeds, and the shaded area represents the standard deviation. The damping is negative due to the sign convention in MuJoCo.

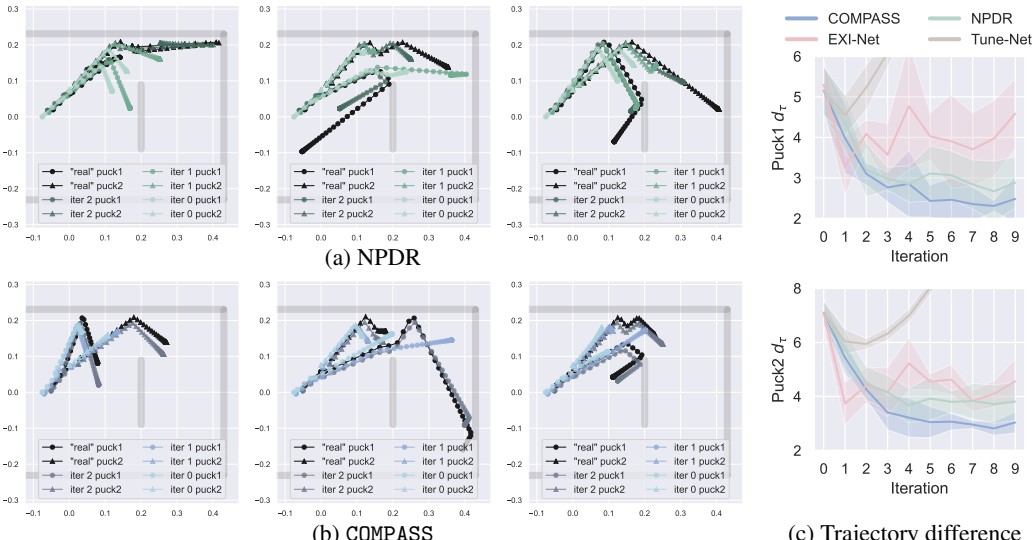

Figure 5: (a) and (b) visualize sampled trajectory-aligning results using NPDR and COMPASS, respectively. The overlapping of trajectories and the obstacle are due to the camera bias. (c) shows the mean trajectory differences throughout 10 iterations, evaluated with 5 random seeds and 10 pairs of rollouts per seed.

difference because we initialized the position of puck1 and puck2 in such a way that they almost only collided with the right wall. Though occasionally pucks did collide with the front wall or the obstacle, the speed at contacting is usually low and has minimal effects on the trajectory; therefore COMPASS placed low attention to such causality. Compared with the baseline methods, our proposed method provides unique interpretability that could guide the sim-to-real transfer in other systems.

The optimization process of 8 environment parameters with discovered causality is shown in Figure 4. We observe that COMPASS optimizes the environment parameters to approach the ground truth gradually, while baseline methods struggle to converge, especially for damping. Since Tune-Net trains a regression model to predict the difference between the current and the target set of environment parameters given trajectories, it does not work well when most of the environment parameters have negligible effects on the trajectories. Note that our method does NOT necessarily converge to the ground-truth environment parameters; rather, it aims to find a combination of environmental parameters to minimize the trajectory difference (more discussion in Section 5). The trajectory-aligning results are shown in Figure. 5(a)(b). It is observed that our method outperforms all the baselines in terms of the trajectory difference using the same number of "real" and simulation rollouts.

Table 1: Trajectory difference (averaged between Puck1 and Puck2) and agents' performance in the real environment. "±" represents the standard deviation. We evaluate the results using 5 policies generated from independent runs and collect 10 trajectories for each run.

| | Low fan speed | | | High fan speed | | |
| | Nominal | NPDR | COMPASS | Nominal | NPDR | COMPASS |
|---|---|---|---|---|---|---|
| Trajectory difference min (↓) | 3.68 ± 0.07 | 2.81 ± 0.16 | **2.37 ± 0.10** | 2.23 ± 0.37 | 3.05 ± 0.46 | **1.41 ± 0.25** |
| Trajectory difference max (↓) | 10.77 ± 0.05 | 7.34 ± 1.41 | **5.71 ± 0.23** | 9.84 ± 0.56 | 10.36 ± 0.82 | **8.17 ± 1.34** |
| Trajectory difference mean (↓) | 7.60 ± 0.03 | 5.18 ± 0.77 | **4.02 ± 0.07** | 6.07 ± 0.40 | 5.63 ± 0.5 | **3.97 ± 0.45** |
| Puck2 final dist. to goal center (↓) | 0.35 ± 0.04 | 0.18 ± 0.05 | **0.12 ± 0.03** | 0.29 ± 0.09 | 0.15 ± 0.07 | **0.13 ± 0.02** |
| Success rate (↑) | 0.00 ± 0.00 | 0.39 ± 0.33 | **0.80 ± 0.09** | 0.20 ± 0.07 | 0.47 ± 0.41 | **0.75 ± 0.22** |

### 4.3 Sim-to-real with policy optimization in the loop

In this experiment, we first trained the agent in the initial simulation environment parameters with Soft Actor-Critic (SAC) [48]. Then, we applied COMPASS and the best-performing baseline in the sim-to-sim experiment, NPDR, to update the environment parameters and retrained the agent in the new simulation environment parameters. Finally, we deployed the agent in the real environment and reported the evaluation statistics. We used a fixed set of real trajectories instead of collecting new ones in every iteration. We effectively set up two real environments by powering the electric fan at different speeds, referred to as *low fan speed* and *high fan speed*.

Upon inspecting Table. 1, we first observe that (i) COMPASS consistently outperforms the nominal simulator and the NPDR baseline in terms of trajectory difference and success rate. Notably, COMPASS improved success rate by 105.1% and 59.6% compared with NPDR in the low and high fan speed settings, respectively. We hypothesize that COMPASS is more robust to the noisy and unmodeled dynamics in the real environment owing to the sparse causal model learned during the model learning and parameter optimization process. We also observe that (ii) the agent's real-world performance positively correlates with the trajectory alignment performance. This is as expected given that previous works have proved that the policy performance degradation is bounded by the difference in transition distributions between two systems [39]. Similar patterns are observed in the supplementary experiments as well (Appendix E).

## 5 Discussion and Conclusion

In conclusion, COMPASS is a novel causality-guided framework to identify simulation environment parameters that minimize the sim-to-real gap. It has three salient features. Firstly, COMPASS requires less domain knowledge of the randomized environment parameters, enabling a more automated process for sim-to-real transfer. Secondly, COMPASS learns an interpretable causal structure, providing better generalization during environment parameter optimization and robustness against observational noise in real rollouts. Lastly, COMPASS employs a fully differentiable model to update the environment parameters, which mitigates the efficiency issue of the existing sampling-based methods. Through both simulation and real-world experiments, we verify that our proposed method outperforms the existing gradient-free and gradient-based parameter estimation methods in terms of trajectory alignment accuracy and the agent's success rate, while offering interpretability.

**Limitations.** With all the advantages of COMPASS discussed, some limitations of our method also suggest directions for future work. Similar to the existing gradient-based methods [34, 36, 35, 21], COMPASS could converge to local minima since the identifiable set of parameters may be coupled [49, 50], which would result in multiple local minima in the parameter space. Indeed, if the purpose is to find a specific combination of parameters that minimize the sim-to-real gap, it becomes less important whether it converges to the global optimum or not [34]. In addition, COMPASS finds a single combination of environment parameters rather than a distribution of them. Nevertheless, our method can maintain several particles of environment parameters as an empirical distribution [51, 52, 53, 54, 55] without extensive modifications to the core algorithm.

**Acknowledgments**

The authors gratefully acknowledge the support from the National Science Foundation under grants CNS-2047454.

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

# A  Model details

In a standard fully-connected multilayer perceptron (MLP), the input is treated as a whole and input into the first linear layer. It blends all information into the feature of subsequent layers, making it difficult to separate the cause and effects. To highlight the difference between our model and traditional MLP, we plot the detailed model architecture of the COMPASS in Figure. 6.

More specifically, we design an encoder-decoder structure in $f$, with $\mathcal{G}$ as a linear transformation applied to the intermediate features. First, the encoder operates on each dimension of $\epsilon$ independently to generate features $z_{\epsilon} \in \mathbb{R}^{|\mathcal{E}| \times d_z}$. Then the causal graph is multiplied by the features to generate the inputs for the decoder, i.e., $g_{\epsilon} = z_{\epsilon}^T \mathcal{G} \in \mathbb{R}^{d_z \times K}$, where $d_z$ is the dimension of the feature. Similarly, the action sequence $a$ is passed through the encoder and transformation to produce the feature of the action sequence $g_{a} \in \mathbb{R}^{d_z \times K}$. Finally, $g_{\epsilon} + g_{a}$ is passed through the decoder to output the prediction $\hat{d}_{\tau}$.

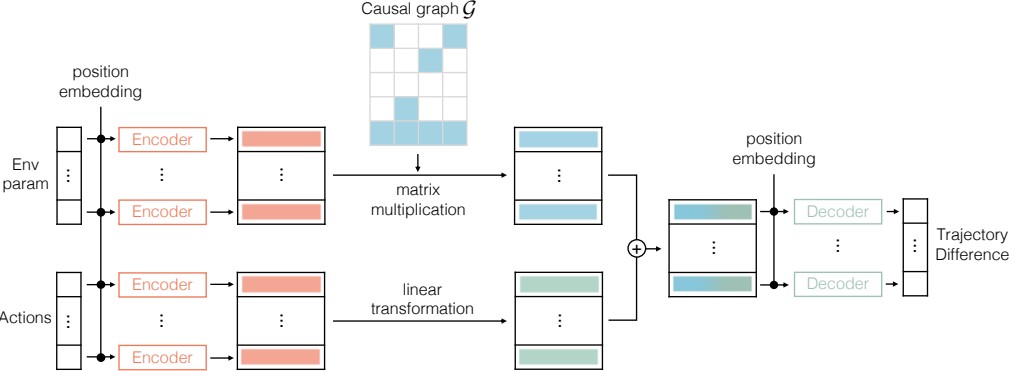

Figure 6: COMPASS model architecture.

# B  Experimental Details

### B.1  Experimental Configurations for Robot Simulation Setup

A simulator was developed to mimic a simulated air hockey table with dimensions matching the real-world setup using robosuite [18]. This allowed us to gather rollout trajectories and training data within the simulated environment. The specific dimensions of the objects within the simulation, such as the table size, are detailed in Table 2. Additionally, for the sim-to-sim experiment configuration, default simulation parameters and the ground-truth simulated real environment parameters are presented in Table 3.

**State and action space.** The observation space for the RL agent comprises 6 dimensions. It includes the initial position of puck1 (3D), and the initial position of puck2 (3D). The RL agent's action space consists of 4 dimensions: the initial position of the pusher in the x-direction, the initial position of the pusher in the y-direction, the shooting angle of the pusher, and the pushing velocity of the pusher. We fixed the initial position of puck1 and puck2. Note that the shooting angle is relative to the line connecting the center of the pusher and puck1.

**Reward function.** The reward is calculated as $-10 \times \|[x_{puck1}, y_{puck1}, z_{puck1}] - [x_{goal}, y_{goal}, z_{goal}]\|_2$. To provide additional incentive for reaching the goal, the distance penalty term is divided by 2. Furthermore, a terminal reward is given if the hockey stays within the success region at the last time step. It is important to mention that the reward is not accumulated throughout the horizon. Instead, it only considers the final Euclidean distance between puck2 and the goal center. For further numerical details and specifications, please refer to Table 4.

Table 2: Mujoco Simulation Environment Setup

|  | X $(m)$ | Y $(m)$ | Z $(m)$ | Radius$(m)$ |
|---|---|---|---|---|
| Air hockey table | 0.0 | 0.0 | 0.8 | [0.45, 0.9, 0.035] |
| Puck1 | -0.15 | 0.0 | 0.8 | 0.0255 |
| Puck2 | -0.075 | -0.075 | 0.8 | 0.0255 |
| Goal point | 0.43 | 0.0 | 0.8 | 0.15 |
| Obstacle bar | 0.1 | 0.0 | 0.8 | [0.025, 0.18, 0.025] |

Table 3: Sim-to-Sim Env Parameters

| Env param | Default Simulation Env Parameters | Simulated "Real" Env Parameters |
|---|---|---|
| pusher@actuation@vel_discount | 0.75 | 0.85 |
| pusher@dyna@damping | -10.0 | -6.0 |
| puck1@dyna@damping | -10.0 | -6.0 |
| puck1@dyna@friction_sliding | 0.05 | 0.04 |
| puck2@dyna@damping | -10.0 | -6.0 |
| puck2@dyna@friction_sliding | 0.05 | 0.03 |
| front_wall, back_wall, left_wall, right_wall, obstacle@dyna@damping | -10.0 | -6.0 |
| env@camera@bias_x | 0.0 | +0.03 |
| env@camera@bias_y | 0.0 | -0.02 |

## B.2 Experimental Configurations for Real Robot Setup

This is a top-down view of the mini air hockey table we used to collect real trajectories. The dimensional attributes of each component are annotated in Figure 7, while green dashed lines distinctly demarcate the designated goal area. We used Kinova Gen 3 robot platform and installed a top-down Intel RealSense D345f RGB camera to track the position of puck1 and puck2.

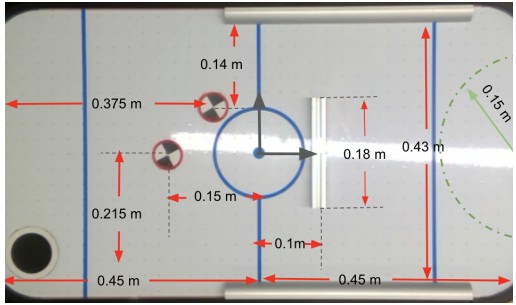

Figure 7: Top-Down View of Table Air Hockey.

## C Implementation Details

We reproduced the baseline implementation EXI-Net [36], NPDR [20] and Tune-Net [34] based on the papers and released code base. We utilized software packages such as PyTorch [56], sbi [57], StableBaseline3 [58], and OpenCV [59]. We report the hyperparameters used for all algorithms,

Table 4: Simulation Environment Setup

| Simulation Parameters | |
|---|---|
| Action space | low: [-0.24, 0.065, -0.157, 0.3] high: [-0.21, 0.085, 0.157, 0.5] |
| Observation space | low: [-inf] — high:[inf] |
| Terminal reward | 2.25 |
| Simulation horizon | 50 time steps |
| Simulation timestep | 0.05 $s$ |

Table 5: Soft Actor-Critic Hyperparameters

| Parameters Name | Values |
|---|---|
| learning_rate | 3e-4 |
| gradient_steps | 32 |
| batch_size | 32 |
| train_freq | 8 |
| ent_coef | 0.005 |
| net_arch | [32, 32] |
| policy | "MlpPolicy" |
| env_number | 64 |
| buffer_size | 1,000,000 |
| learning_starts | 100 |
| tau | 0.005 |
| gamma | 0.99 |
| action_noise | None |
| stats_window_size | 100 |

including the proposed `COMPASS` method, in Table 6, 7, 8, and 9, which provide a comprehensive list of the parameters we utilized to reproduce the results.

**Soft Actor-Critic hyperparameters.** We use SAC implementation in StableBaseline3 [58] to train the RL agents. The training hyperparameter is shown in Table 5.

Table 6: `COMPASS` hyperparameters

| Description | value | variable_name |
|---|---|---|
| Shared Hyperparameters | | |
| Number of iterations | 10 | n_round |
| Retrain in each iteration (if False, keep using the model trained in the first iteration) | True | retrain_from_scratch |
| Number of rollouts in each iteration | 640 | n_samples_per_round |
| Number of command actions in each iteration | 10 | n_cmd_action |
| Number of epochs | 4000 | n_epochs |
| Batch size | 64 | batch_size |
| Learning rate | 0.001 | learning_rate |
| Algorithm-Specific Hyperparameters | | |
| Network encoder dimension | 32 | emb_dim |
| Network hidden dimension | [256, 256] | hidden_dim |
| Causal dimension | 32 | causal_dim |
| Sparsity weight of the loss function | 0.003 | sparse_weight |
| Sparsity weight discount | 0.5 | sw_discount |
| Loss function | MSE + Sparsity | loss_function |
| Optimizer | Adam | optimizer |

Table 7: EXI-Net hyperparameters

| Description | value | variable_name |
|---|---|---|
| Shared Hyperparameters | | |
| Number of iterations | 10 | n_round |
| Retrain in each iteration (if False, keep using the model trained in the first iteration) | True | retrain_from_scratch |
| Number of rollouts in each iteration | 640 | n_samples_per_round |
| Number of command actions in each iteration | 10 | n_cmd_action |
| Number of epochs | 4000 | n_epochs |
| Batch size | 64 | batch_size |
| Learning rate | 0.001 | learning_rate |
| Algorithm-Specific Hyperparameters | | |
| Network hidden dimension | [256, 256] | hidden_dim |
| Loss function | MSE | loss_function |
| Optimizer | Adam | optimizer |

Table 8: NPDR hyperparameters

| Description | value | variable_name |
|---|---|---|
| Shared Hyperparameters | | |
| Number of iterations | 10 | n_round |
| Retrain in each iteration (if False, keep using the model trained in the first iteration) | True | retrain_from_scratch |
| Number of rollouts in each iteration | 640 | n_samples_per_round |
| Number of command actions in each iteration | 10 | n_cmd_action |
| Algorithm-Specific Hyperparameters | | |
| Prior distribution type | Uniform | prior |
| Inference model type | maf | inf_model |
| Embedding net type | LSTM | embedding_struct |
| Embedding downsampling factor | 2 | downsampling_factor |
| Posterior hidden features | 100 | hidden_features |
| Posterior number of transforms | 10 | num_transforms |
| Normalize posterior | False | normalize_posterior |
| Density estimator training epochs | 50 | num_epochs |
| Density estimator training rate | 3e-4 | learning_rate |
| Early stop epochs once posterior converge | 20 | stop_after_epochs |
| Use combined loss for posterior training | True | use_combined_loss |
| Discard prior samples | False | discard_prior_samples |
| Sampling method | MCMC | sample_with |
| MCMC thinning factor | 2 | thin |

Table 9: Tune-Net hyperparameters

| Description | value | variable_name |
|---|---|---|
| Shared Hyperparameters | | |
| Number of iterations | 1 | n_round |
| Retrain in each iteration (if False, keep using the model trained in the first iteration) | False | retrain_from_scratch |
| Number of rollouts in each iteration | 6400 | n_samples_per_round |
| Number of command actions in each iteration | 10 | n_cmd_action |
| Number of epochs | 4000 | n_epochs |
| Batch size | 64 | batch_size |
| Learning rate | 0.001 | learning_rate |
| Algorithm-Specific Hyperparameters | | |
| Network input dimension (Pair of Trajectory and Action dimension) | (2, 304) | (dim_pair, dim_state) |
| Network output dimension (Tunable env param dimension) | 64 | dim_zeta |
| Env param update iteration | 10 | K |
| Network hidden dimension | [256, 256] | hidden_dim |
| Loss function | MSE | loss_fn |
| Optimizer | Adam | optimizer |

# D Supplementary Ablation Study for Air-Hockey Experiment

## D.1 Different Real Rollout Size $N$ and Sim Rollout Size $M$

We investigated the effect of the real rollout size $N$ and simulation rollout size $M$ by testing values of $N = 5, 10, 20$ at $M = 64$ and $M = 32, 64, 128$ at $N = 10$. The optimization processes for environment parameters, given these sizes, are illustrated in Figure 8 and 9. Notably, for a given $N, M$, COMPASS demonstrates superior convergence in comparison to NPDR.

Further insights are offered in Figure 10a and 10c, which depicts the trajectory differences over iterative steps, and Figure 10b and 10d, showing the final trajectory discrepancies for different $N, M$ values. These visualizations highlight that as the rollout budget $N, M$ diminishes, the performance gap between COMPASS and other benchmark methods widens. This can be attributed to the fully-differentiable nature of COMPASS. Reinforcing our key contributions, COMPASS consistently outperforms the baselines, maintaining superiority under identical counts of real and simulation rollouts.

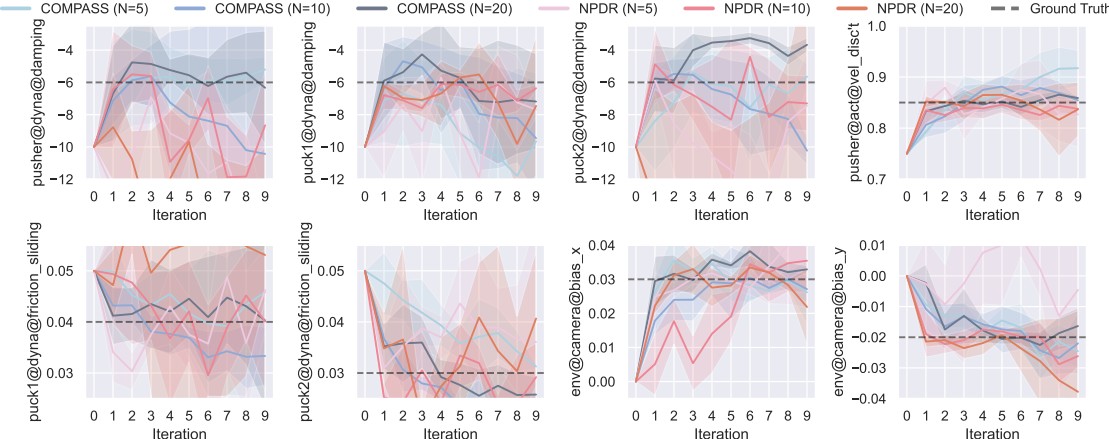

Figure 8: Environment parameters optimization with different real rollout sizes $N$. We show 8 parameters with discovered causality (as shown in Figure 3). The solid lines represent the mean value across 3 random seeds, and the shaded area represents the standard deviation.

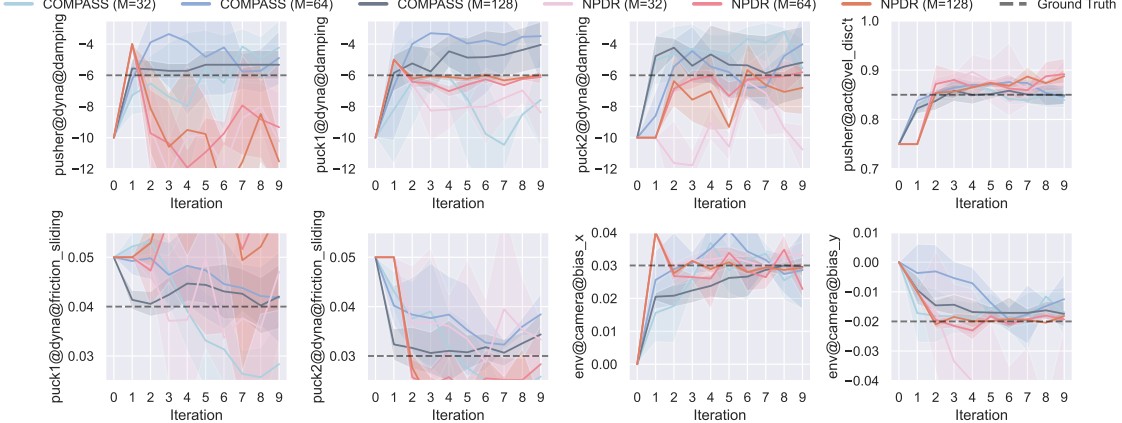

Figure 9: Environment parameters optimization with different sim rollout sizes $M$. We show 8 parameters with discovered causality (as shown in Figure 3). The solid lines represent the mean value across 3 random seeds, and the shaded area represents the standard deviation.

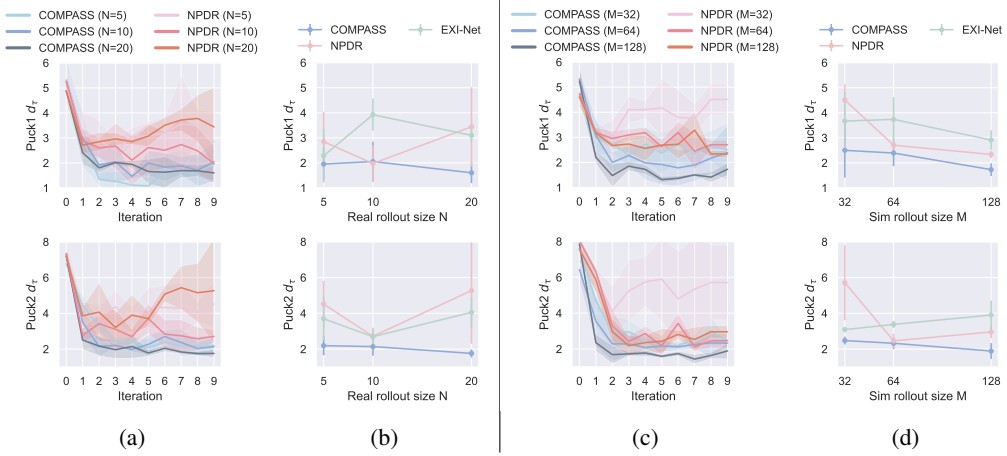

Figure 10: (a)(b) Trajectory difference for different real rollout size $N$. (c)(d) Trajectory difference different sim rollout size $M$.

## D.2 Different Initial Environment Parameters

To investigate the sensitivity of COMPASS to initialization, we conducted experiments with randomly initialized environment parameters. The environment parameter optimization processes are illustrated in Figure.12, while Figure.11 presents the trajectory discrepancies.

A key observation from both figures is that the starting point—i.e., the parameter initialization—does influence the complexity of aligning the trajectories. Despite these variances in initialization, COMPASS consistently demonstrates its capability to minimize the trajectory difference across different initial conditions.

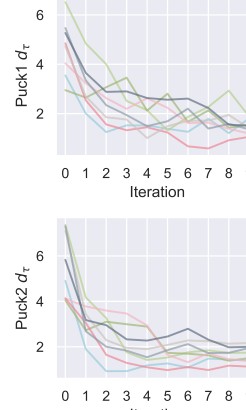

Figure 11: Trajectory difference.

## D.3 Different Sparsity Weight $\lambda$

The learned causal graph parameters, denoted as $\psi$, are illustrated in Figure. 13 across various sparsity weights: $\lambda = 0.001, 0.005, 0.01$. In the absence of regularization (specifically when $\lambda = 0$), the graph tends to be denser, devoid of any constraints to minimize edge count. As $\lambda$ increases, the

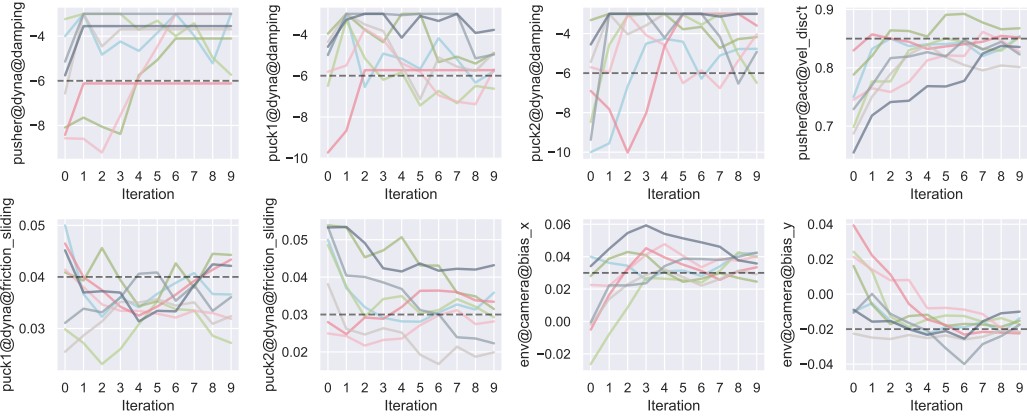

Figure 12: Environment parameters optimization with randomized initialization. We only show one run per initialization.

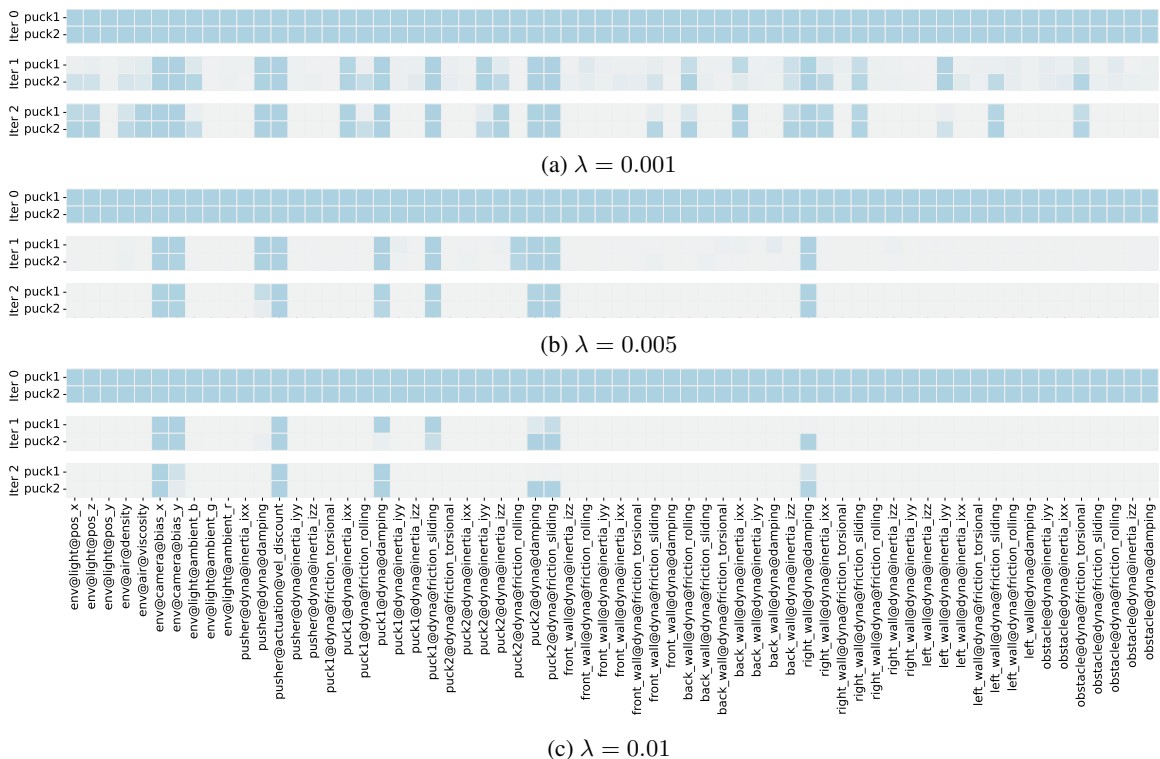

(a) $\lambda = 0.001$

(b) $\lambda = 0.005$

(c) $\lambda = 0.01$

Figure 13: Learned causal graph parameters $\psi$ with different sparsity weight $\lambda$. Darker colors present values closer to 1. We use parameter notations in the format of *object@param_type@param*.

graph becomes progressively sparser. However, it still maintains the most influential edges; omitting them would significantly elevate the prediction error.

# E    Sim-to-Real Double-Bouncing-Ball Experiment

## E.1    Double-bouncing-ball experimental setup

Figure 14 illustrates the experimental setup. Building upon the experimental design presented by Allevato et al. [34], we've raised the bar by allowing the ball to bounce twice, rather than just once, before landing in the goal basket. The goal basket has a diameter of 13.0 cm, and the ball measures 6.8 cm in diameter. Successful task completion is characterized by the ball's accurate entry into the hoop from above.

The task's intricacy hinges on achieving a seamless alignment between the real-world dynamics and the simulation. By releasing the ball at a specific height, it needs to bounce first off inclined plate 1, followed by plate 2, and finally enter into the goal basket. The ball's motion is constrained within a 2-D plane with properly aligned plates. As such, the action space is represented by a scalar - the ball's release height. Simultaneously, the state space corresponds to the ball's 2-D positional coordinates within this plane ($K = 1$). This experiment encompasses 82 environment parameters ($|\mathcal{E}| = 82$).

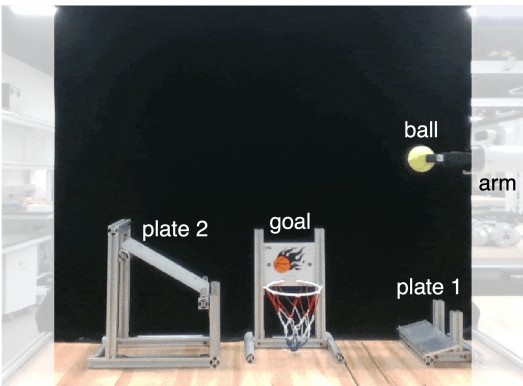

Figure 14: Double-bouncing-ball setup.

## E.2    Learned causal graph

Figure 15 depicts the learned causal graph parameter, denoted as $\psi$. Starting from a fully-connected causal graph, COMPASS efficiently narrows down the parameter search space from 82 dimensions to a mere 4 dimensions. Specifically, these are *ball@dyna@damping*, *ball@dyna@mass*, *plate1@dyna@damping*, and *plate2@dyna@damping*. Remarkably, this refinement is achieved within just two iterations.

## E.3    Sim-to-real trajectory alignment results

As depicted in Figure 16, it's evident that COMPASS effectively aligns simulated trajectories closely with their real-world counterparts. Given the extensive scale of environment parameters in this con-

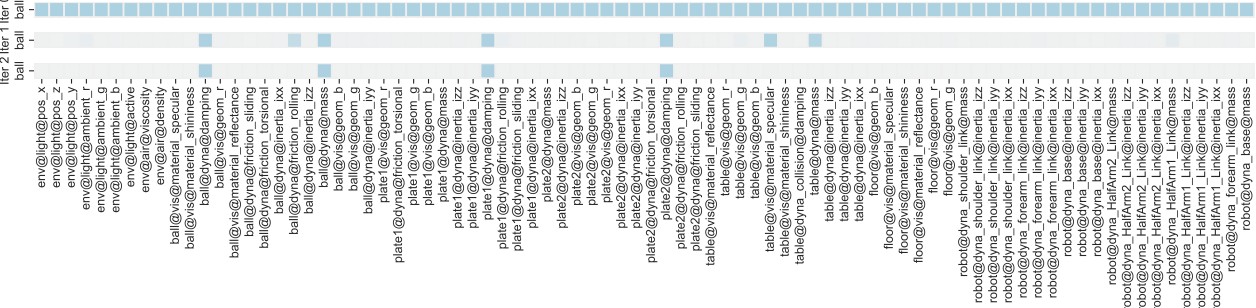

Figure 15: Learned causal graph parameters $\psi$ for double-bouncing-ball experiments. Darker colors present values closer to 1. We use parameter notations in the format of *object@param_type@param*.

text, sampling-based techniques like NPDR become notably inefficient. This is primarily because they necessitate an immense sample size to learn a posterior distribution accurately.

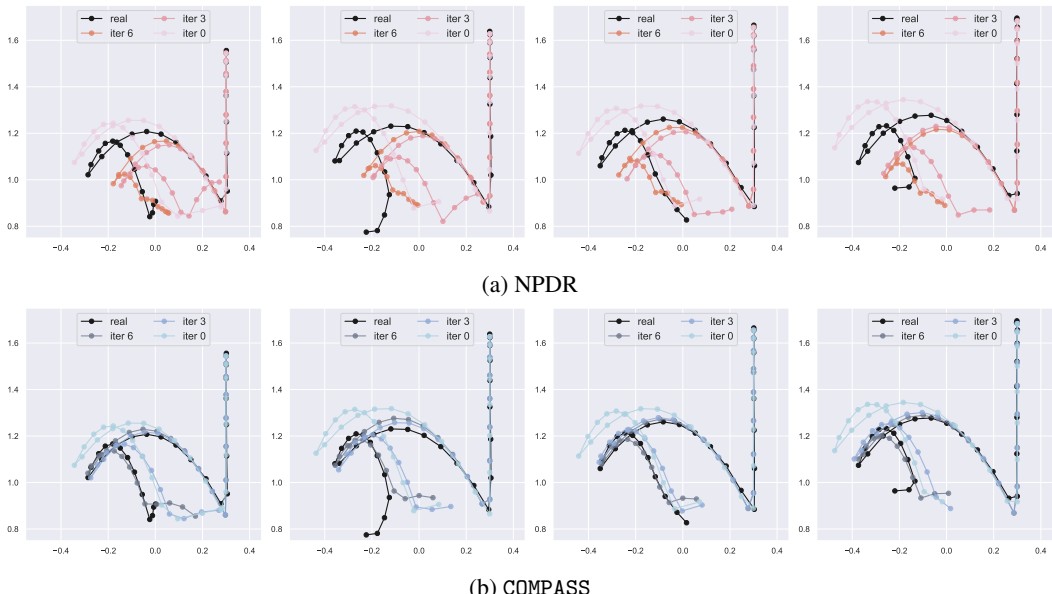

(a) NPDR

(b) COMPASS

Figure 16: Visualization of sim-to-real trajectory alignment.

## E.4 Real-world agent performance

Table 10 presents the real-world performance of agents after policy optimization in the adjusted environment, compared with nominal agents (those trained in the original environments). Consistent with the trajectory alignment outcomes, agents trained using COMPASS demonstrate superior success rates compared to those trained using NPDR and the nominal approach.

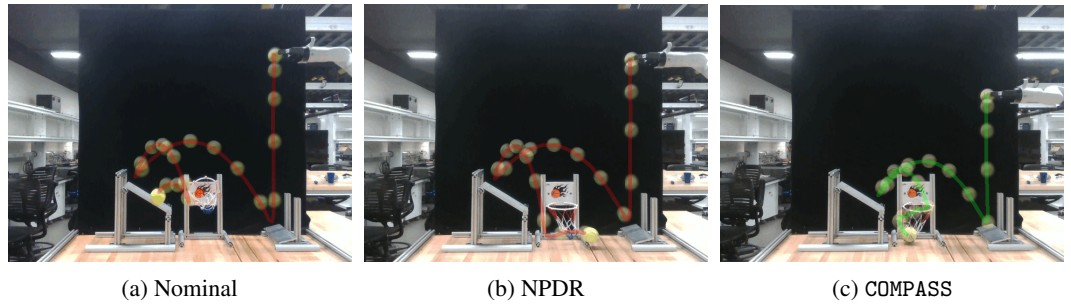

(a) Nominal          (b) NPDR          (c) COMPASS

Figure 17: Real-world policy rollouts.

Table 10: Agents' performance in the real environment. "±" represents the standard deviation. We evaluate the results using 5 policies generated from independent runs and collect 10 trajectories for each run.

|  | Nominal | NPDR | COMPASS |
|---|---|---|---|
| Success rate ($\uparrow$) | $0.30 \pm 0.14$ | $0.58 \pm 0.15$ | $\mathbf{0.86 \pm 0.10}$ |

# F  Sim-to-Sim Push-I Experiment

## F.1  Experimental setup

We conducted further evaluations using an experiment inspired by the Push-T experiment described by Chi et al. [60]. In this experiment, illustrated in Figure. 18, the agent controls the robot arm to strategically move a slender I-shaped cube towards a specific position and alignment. The task spans a horizon of $T = 30$. The state space comprises the 3-D coordinates and orientation of the cube, the target, and the end effector. The action space is defined by the 3-D velocity of the end effector and the gripper's action (either open or close). The states of interest are factorized into the position, roll, pitch, and yaw of the cube ($K = 4$). This setup involves 67 environment parameters ($|\mathcal{E}| = 67$).

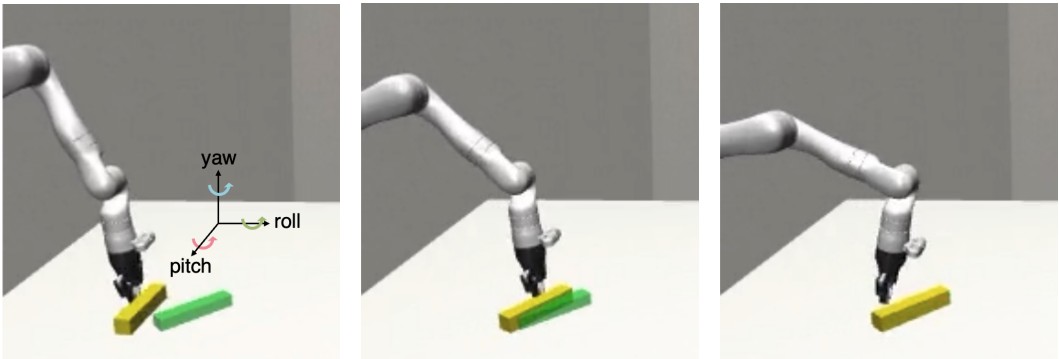

Figure 18: Push-I task. The green cube represents the target.

## F.2  Learned causal graph

The learned causal graph parameter is shown in Figure. 19. Starting from a fully-connected causal graph, COMPASS efficiently narrows down the parameter search space from 67 dimensions to only 3 dimensions within 2 iterations. Specifically, they are *cube@dyna@mass*, *cube@dyna@damping*, *cube@dyna@friction_sliding*.

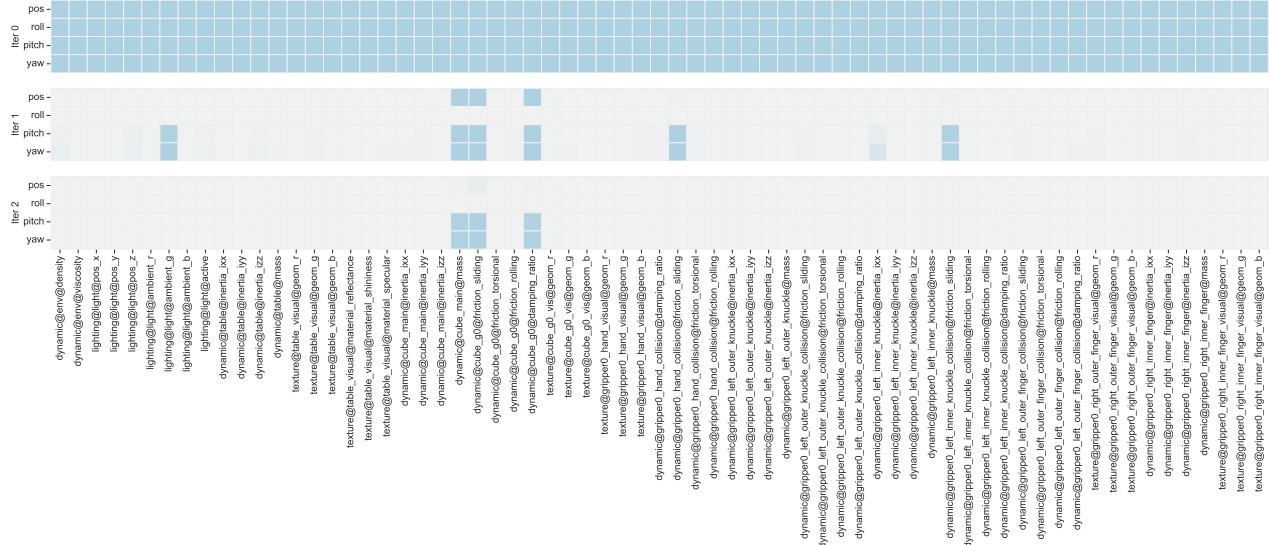

Figure 19: Learned causal graph parameters $\psi$ for Push-I experiments.

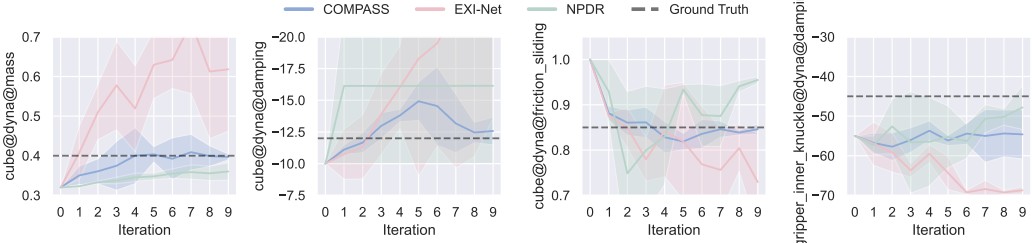

Figure 20: Environment parameters optimization. The solid lines represent the mean value across 3 random seeds, and the shaded area represents the standard deviation.

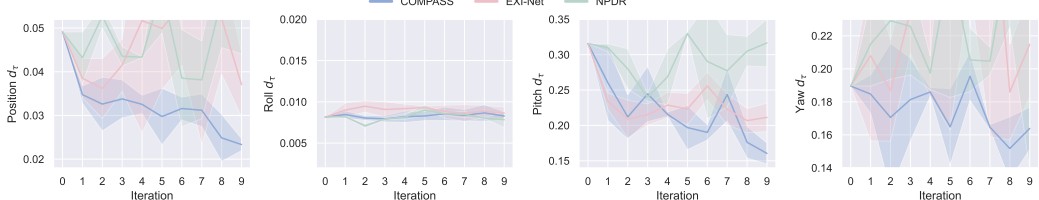

Figure 21: Trajectory difference optimization results. The solid lines represent the mean value across 3 random seeds, and the shaded area represents the standard deviation.

### F.3 Environment parameter optimization and trajectory alignment

Figure.20 depicts the optimization of environment parameters across iterations. We observe that COMPASS optimizes the environment parameters to approach the ground truth gradually, while baseline methods struggle to converge. The trajectory difference over the iterations is presented in Figure. 21, while the visualization of trajectory alignment is shown in Figure. 22, 23, 24, 25. At first glance, the trajectory difference in the roll angle doesn't seem to show much reduction. However, a closer examination of Figure.22 reveals that the roll angle has minimal changes during the whole trajectory regardless of the environment parameters. Given the elongated shape of the cube (illustrated in Figure. 18), it's reasonable to see only minor variations in this direction. Overall, these figures highlight COMPASS's efficiency in aligning the simulator with the real world.

### F.4 Sim-to-sim agent performance

Table 10 presents the "real" performance of agents after policy optimization in the adjusted environment, compared with nominal agents (those trained in the original environments). Consistent with the trajectory alignment outcomes, agents trained using COMPASS demonstrate higher success rates compared to those trained using NPDR and the nominal approach.

Table 11: Agents' performance in the "real" environment. "±" represents the standard deviation. We evaluate the results using 3 policies generated from independent runs and collect 10 trajectories for each run.

|  | Nominal | NPDR | EXI-Net | COMPASS |
|---|---|---|---|---|
| Success rate (↑) | $0.10 \pm 0.22$ | $0.40 \pm 0.21$ | $0.50 \pm 0.22$ | $\mathbf{0.70 \pm 0.28}$ |

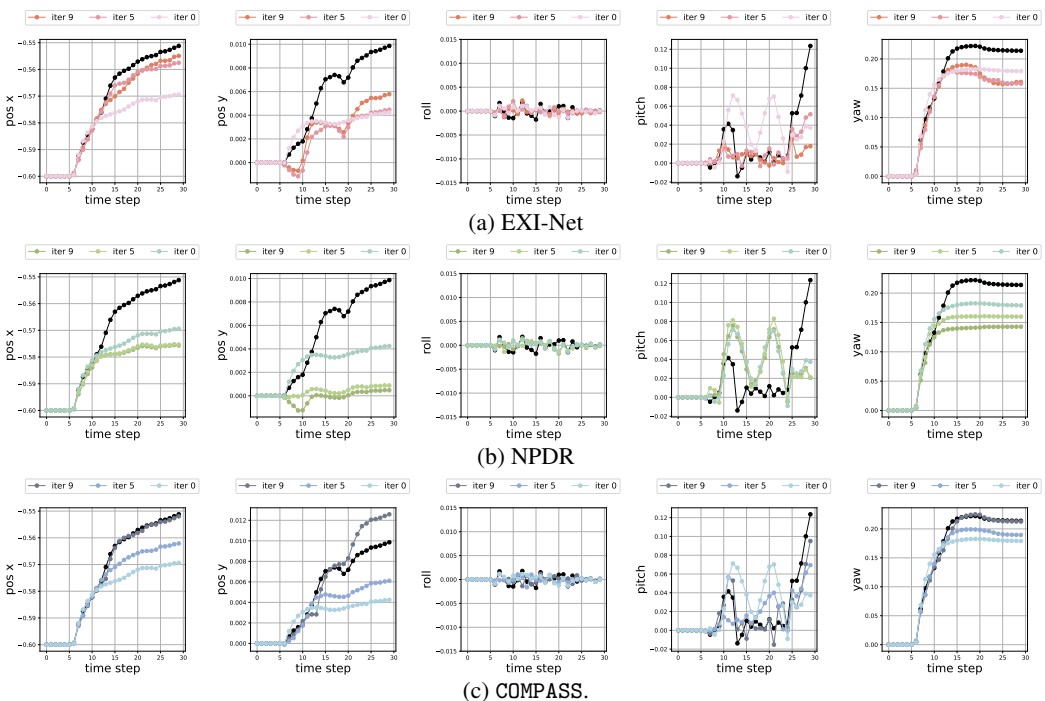

Figure 22: Trajectory alignment results of EXI-Net, NPDR, and COMPASS.

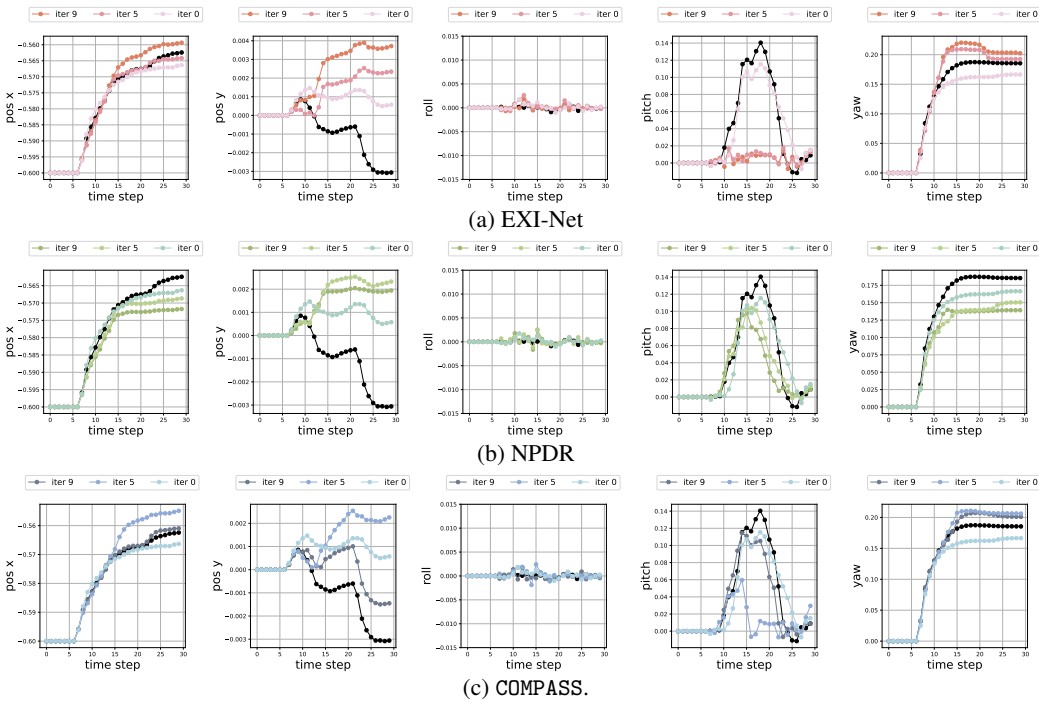

Figure 23: Trajectory alignment results of EXI-Net, NPDR, and COMPASS.

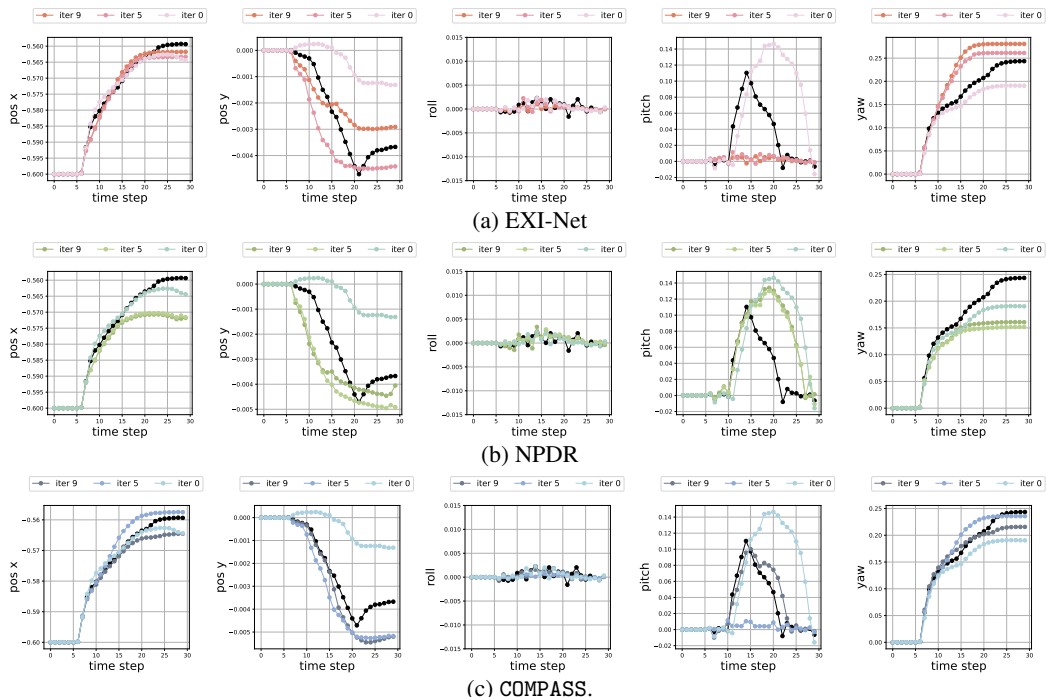

Figure 24: Trajectory alignment results of EXI-Net, NPDR, and COMPASS.

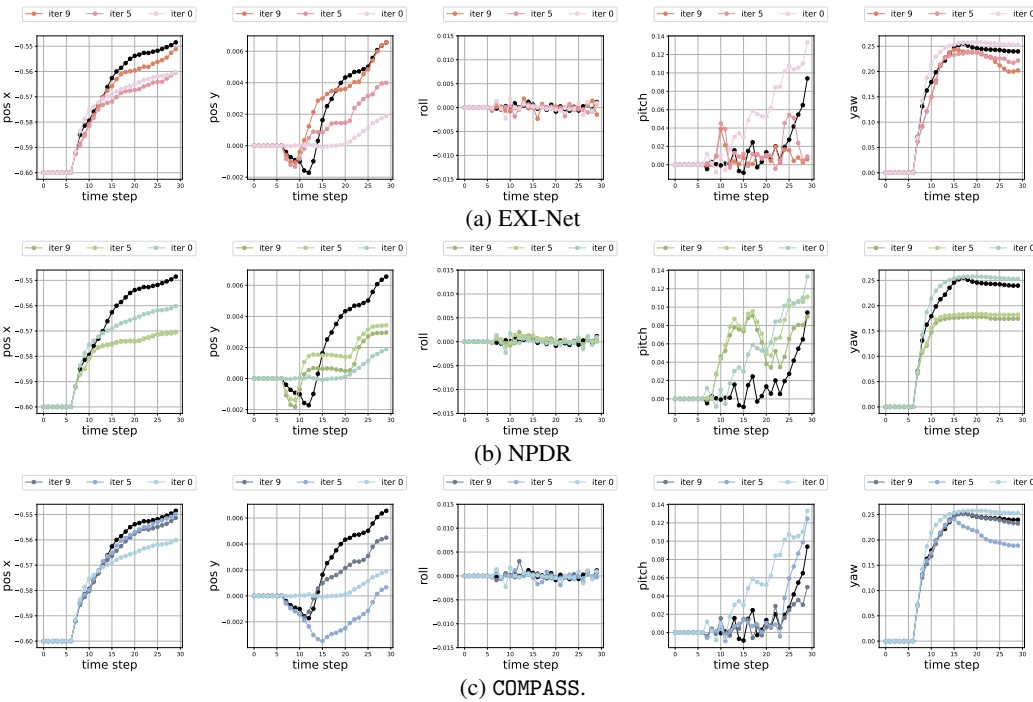

Figure 25: Trajectory alignment results of EXI-Net, NPDR, and COMPASS.

## G  Additional Literature Review

**Adaptive Policy in Locomotion and Manipulation.** The challenge of sim-to-real transfer has been central to the field of locomotion tasks, and it has recently demonstrated remarkable success [61, 62, 63, 64, 65]. Rapid Motor Adaptation (RMA) [62] proposed a solution to bridge the sim-to-real gap by effectively learning the relationship between dynamic-affecting parameters and historical contexts. More recently, Kumar et al. [63] introduced Adapting-RMA (A-RMA) to further refine the base policy of RMA using model-free reinforcement learning (RL) techniques. Typically, RMA-based methods approach the sim-to-real challenge as a generalization problem. They tend to assume an appropriate range and set of parameters that influence testing performance, along with a sizable randomized training budget, to ensure successful operation. These assumptions present inherent challenges due to the requisite domain expertise and training time. In manipulation, Liu et al. [66] approached the adaptive policy from a continual RL perspective, cultivating a policy for each group of tasks rather than an individual task to solve unseen tasks in seen groups in a zero-shot manner. In contrast, this paper focuses on aligning simulators with real-world dynamics. Our approach involves the automated identification of simulation environment parameters that minimize the sim-to-real dynamics gap. While there are similar studies, such as the work by Mozian et al. [65], which on searching for the environment parameter distributions that are challenging yet not excessively adversarial to learn, our emphasis is on sim-to-real applications with novel causality-based system identification.

