# OpenReview forum: "What Went Wrong? Closing the Sim-to-Real Gap via Differentiable Causal Discovery"
_robot-learning.org/CoRL/2023/Conference — CoRL 2023 Poster_

### Official Review · Reviewer_Ltn4 · 2023-07-12

**Confidence:** 4
**Originality:** Very Good
**Technical Quality:** Very Good
**Clarity Of Presentation:** Very Good
**Impact:** 4

**Recommendation:**

Strong Accept: I recommend accepting the paper and will argue for my recommendation even if other reviewers hold a different opinion.

**Review:**

This work introduces `COMPASS`, a method designed to align simulators more closely with the real world by discovering the causality
between environment parameters and the sim-to-real gap. COMPASS achieves this by learning a differentiable mapping from simulation environment parameters to differences between simulated and real-world robot-object interaction trajectories. This mapping process is governed by a concurrently learned causal graph.

After the causal model is established, COMPASS employs back-propagation to optimize simulation environment parameters
in an end-to-end manner. The causal graph helps the model by enhancing interpretability, pruning the parameter search space, and thereby improving the efficiency of domain randomization and scalability.

The causal graph focuses on the influence of environment parameters ($\epsilon$) on the trajectory difference ($d\tau$).
It is structured with predefined directed edges and represented by a binary adjacency matrix of size $|E| \times K$,
where 1 and 0 indicate the existence and absence of an edge, respectively.

Inspired by previous works, the authors formulate combinatorial graph learning into a continuous optimization
problem, making the optimization of `G` differentiable. They designed a pipeline where elements
 of the graph `G` are sampled from a `Gumbel-Softmax` distribution, parameterized by $\psi$.

The causal graph is denoted as $G \psi$, with all elements (i, j) initialized to ones to ensure the graph is fully
connected at the start. In addition to the causal graph, the authors underscore the need for a parameterized model to accurately
represent the influence of causes on effects. They design an encoder-decoder structure (`f`)
in which the causal graph (`G`) is applied as a linear transformation to intermediate features.

The encoder processes each dimension of the environment parameters independently to generate
features $z_\epsilon$. The causal graph then multiplies these features to create inputs for the decoder $(g\epsilon = z\epsilon T G)$,
where dz is the dimension of the feature. The action sequence (a) follows a similar process through the
encoder and transformation, yielding the feature of the action sequence ($ga$). Finally, the sum of $g\epsilon$ and $ga$ is passed through the decoder to output the predicted trajectory difference ($dˆ \tau$).
This process  captures how changes in environment parameters and action sequences can influence the
difference between simulated and real-world trajectories.

**Experiments**

The robotics task is a `Mini Air Hockey with Obstacle`. To reach the goal, the agent needs to consider pusher-to-puck, puck-to-puck, puck-to-wall collisions, and surface properties of the hockey table. Baselines are `Neural Posterior Domain Randomization (NPDR)`, `EXI-Net` and `Tune-Net`.

- Sim-to-sim trajectory alignment with known target environment parameters
This experiment verifies whether COMPASS can align trajectories between two different envionments.

- Sim-to-real with policy optimization in the loop
For the policy learning, Soft Actor-Critic (SAC) is used and COMPASS is applied to update the environment parameters and retrained the agent in the new simulation environment parameters.



**Quality Of The Limitations Section:**

Limitations are addressed clearly

**Questions For Rebuttal:**

- Missing baseline : I think there are some missing baselines including [1] which you should cite. They are also learning a domain randomization distribution, just with a different methodology. Even though they are not doing sim vs real trajectory alignment.

- How does COMPASS handle high dimensional spaces and the so-called "curse of dimensionality"? Could you further explain how the differentiable model helps in mitigating this issue?

- How does COMPASS interact with the reinforcement learning (RL) components in the system? Does the process of learning the causal graph have any impact on the RL policy optimization? As far as I understood this process is currently disjoint. I'm wondering if the authors tried learning the parameters and the policy at the same time.

- How does the regularization term encourage the elimination of irrelevant environment parameters in the objective function? I did not see any ablation study of the effect of this term.

- I got a little bit lost when reading the encoding part where the method requires the parameters and the action sequence. Are these actions interpreted as interventions ? And are these, so to speak, the correct actions ? The actions that successfully take you to the next desired state?

[1] Learning Domain Randomization Distributions for Training Robust Locomotion Policies - https://ieeexplore.ieee.org/abstract/document/9341019

**Robotics Focus:**

Sufficient demonstration on hardware

**Summary Of Paper:**

The paper discusses the challenges associated with training robot control policies in simulations that
need to transfer to the real world, a problem often termed the simulation-to-real transfer gap.
This gap is often a result of disparities between simulators and the real world, which can produce
inaccurate results. To address these issues, the authors introduce an automated method to tune simulator
parameters. This is achieved by learning the causal relationships between the environment parameters and
the sim-to-real gap. The method works by learning a differentiable mapping from the environment parameters
to the differences between simulated and real-world robot-object trajectories. A causal graph is learned
simultaneously to prune the search space of parameters, provide better interpretability, and improve generalization.

**Summary Of Recommendation:**

This paper introduces COMPASS, a causality-guided framework designed to minimize the simulation-to-real gap. COMPASS identifies and adjusts simulation environment parameters by learning a differentiable mapping from these parameters to the differences between simulated and real-world robot-object trajectories.  The optimization objective includes a mean squared error term for accuracy and a regularization term. The ideas presented here are novel and interesting and the efficacy of the method is demonstrated by performing a sim2real experiment.

---

### Official Review · Reviewer_mBeP · 2023-07-19

**Confidence:** 4
**Originality:** Good
**Technical Quality:** Fair
**Clarity Of Presentation:** Good
**Impact:** 3

**Recommendation:**

Weak Accept: I recommend accepting the paper, but will not argue for my recommendation if the majority of other reviewers have a different opinion.

**Review:**

## Strengths

- The analysis in the sim-to-sim section is well executed.
- The paper is well written and easy to follow.


## Weaknesses

- The biggest weakness lies in the results. There seems to be a missing simpler baseline: using a recurrent based policy in combination with wide-domain randomization; e.g. a recent example is: Kumar et al., Adapting Rapid Motor Adaptation for Bipedal Robots. In general, sim2real for quadruped locomotion has come a long way, and a lot of the related work in that domain seems to be missing. It’s not clear to me why iterating back and forth between sim and real in order to tune the physics simulation is needed.
- Having more than 1 task in the real world would increase the confidence that the findings are applicable to a broad range of tasks.
- The abstract and introduction does not state that the focus of this paper is on dynamics sim-to-real, rather than visual sim-to-real. This is quite imprint to state early, as there are many statements in the first 2 pages that are true for dynamics, but not vision sim2real work; e.g. “…set of tunable parameters is usually manually selected to reduce the search space in a case-by-case manner”.
- There is a contradiction in the motivation for the work; in the first sentence of the introduction, the paper states that “training control policies directly on real robots poses challenges due to potentially unsafe exploration…”; however, the method that the paper presents requires that we roll out a potentially dangerous trajectory in the real world in order to perform the causality-guided domain randomisation. You may want to reconsider this entrance given the presented method.
- Details of the network architecture/hyperparams are missing.

**Quality Of The Limitations Section:**

Limitations are addressed clearly

**Questions For Rebuttal:**

How would this approach compare to more recent work in learning quadruped locomotion?

**Robotics Focus:**

Sufficient demonstration on hardware

**Summary Of Paper:**

The paper proposes COMPASS (Causality between environment parameters and the sim-to-real gap). COMPASS aims to tackle the dynamics sim-to-real gap by learning a differentiable mapping from the simulation environment parameters to the differences between sim and real robot trajectories. This is done by a simultaneously learned casual graph to optimize the simulation parameters so that we end up with a simulation that is as close to the real world as possible. These parameters can then be used in combination with domain randomization to transfer plaices from sim to real.

Experiments are performed in both a sim-to-sim scenario, and a sim-to-real scenario (Mini air hockey with obstacle).

**Summary Of Recommendation:**

Currently the paper is missing crucial baselines that are used in the quadruped sim2real community. To name but a few:

- Hwangbo  et al. Learning Agile and Dynamic Motor Skills for Legged Robots
- Kumar et al., Adapting Rapid Motor Adaptation for Bipedal Robots

---

### Official Review · Reviewer_UgPf · 2023-07-22

**Confidence:** 4
**Originality:** Good
**Technical Quality:** Excellent
**Clarity Of Presentation:** Excellent
**Impact:** 2

**Recommendation:**

Strong Accept: I recommend accepting the paper and will argue for my recommendation even if other reviewers hold a different opinion.

**Review:**

**Quality**
* Overall high technical quality. The experiments clearly assess the effectiveness of COMPASS relative to baseline methods.
* Experiments are described in sufficient detail.
* Insightful analysis assessing the method in a number of different ways. E.g. Fig 3 which highlights how the sparsity of the causal graph changes from iteration to iteration, Fig 4 which assess how well COMPASS finds the correct parameter values vs ground truth and prior methods and finally Fig 5 which assess COMPASS’ performance on the end-to-end trajectory tracking task.
* Well chosen baselines
* Some improvement of COMPASS over the baseline methods
* Nice analysis of the way COMPASS functions (Fig 3)
* Good related work section which clearly situates COMPASS in the broader research context and highlights how COMPASS differs.
* To further improve quality it would be nice to see this evaluated on another task and in a setting where there are more than 2 states in the factorized state space

**Clarity**
* Extremely clear and well written.
* Well motivated.
* One minor suggestion - line 127 - a couple of examples of a factorized state would help a reader build a mental model of how one might factorize a state space.

**Originality**
* The use of a causal graph to prune simulated parameters to optimize is interesting and novel

**Significance**
* It is hard to assess.
* On the one hand this paper presents an original approach to a well known problem, the sim2real gap in robot learning.
* However a competing and simpler strategy is to make use of careful system id combined with extensive domain randomization. The difficulty of working on a variety of heterogeneous hardware often leads to working on a single robot system or a small number of different robots, and under those circumstances a time consuming system id is often worthwhile.
* Given the additional complexity involved in this method it is difficult to assess how widely this might influence the research community. Regardless, I think it is interesting to the community.

**Minor**
* Table 1: Nominal max std deviation overlaps with COMPASS so I think this should be bolded.
* Table 2: High fan speed NPDR has standard deviation overlaps with COMPASS so I think this should be bolded.


**Quality Of The Limitations Section:**

Limitations are addressed clearly

**Questions For Rebuttal:**

In addition to some suggestions and questions in the previous section, please see below for some further questions.
* Table 2: The Puck2 final dist. to goal center for NPDR and COMPASS appear quite similar (0.15 vs 0.13 on average) however the average success rates are quite different (0.47 vs 0.75). Why is this the case?
* What about interactivity between objects?
   * For example, perhaps the accuracy of tracking the 1st puck affects where the 2nd puck is hit and thus indirectly affects the accuracy of the 2nd puck track. Do you see any systematic differences in weight between parameters associated with the 1st vs 2nd puck?
   * If there were many pucks (e.g. 5) would you expect to see COMPASS place successively less weight on the parameters associated with later pucks in the causal chain?
* Why was EXI-NET chosen as the comparison method for visualizing trajectory aligning results in Figure 5 (a) instead of the best baseline, NDPR?
* How sensitive is this method to the initial simulator parameter values? It would be interesting to see a variation of Figure 4 with a variety of different parameter initializations.


**Robotics Focus:**

Sufficient demonstration on hardware

**Summary Of Paper:**

This paper addresses the well known problem of the sim2real gap, that is the difficulty of implementing a simulator such that robot-object trajectories in simulation exactly match the real world. One common method for reducing this gap is to tune various simulated parameters. However this often requires extensive domain knowledge and tends to scale poorly with the number of potential tunable parameters. To address this, the authors propose COMPASS, a method for discovering causal relationships between tunable simulator parameters and the sim2real gap. The causal relationship graph is constructed so as to encourage pruning of relevant parameters, and consequently increases the efficiency of domain randomization and enables targeted environment parameter optimization. The method is evaluated on a mini-air hockey task in which the robot needs to control a puck in order to send a 2nd puck to a goal, avoiding an obstacle.

**Summary Of Recommendation:**

Overall a strong paper. Well written and clearly presented - a pleasure to read. The benefits of COMPASS are demonstrated convincingly and the experimental analysis is insightful. I think it will be relevant and interesting to the research community. All my suggestions are to further improve a paper that I think already passes the acceptance bar.

---

### Official Review · Reviewer_7MJV · 2023-08-01

**Confidence:** 3
**Originality:** Good
**Technical Quality:** Fair
**Clarity Of Presentation:** Very Good
**Impact:** 2

**Recommendation:**

Weak Accept: I recommend accepting the paper, but will not argue for my recommendation if the majority of other reviewers have a different opinion.

**Review:**

The paper is well-written and easy to read. Overall the method of learning a causality graph is interesting as it sheds light on how different parameters affect simulation. Air hockey is a good domain for illustrating and studying the approach as the dynamics are intuitive. On the other hand, for the same reason, the chosen domain does not properly demonstrate the benefit of providing interpretability to complex systems nor improvements over other methods. A substantially more complex (simulated) evaluation would help make the claims of the paper stronger.

The paper proposes Causality-Guided Domain Randomization that selects simulation parameters to be perturbed based on a learned causal graph: dimensions that do not affect simulated trajectories are ignored, and DR is applied to the other dimensions only. However, these ignored dimensions would not affect the simulation anyway, so skipping them seems unnecessary. Instead, choosing the perturbation magnitude ($\delta_r$) is often more challenging and requires manual tuning. Perhaps one could use the graph to automate that too. Moreover, if at iteration $i$, we don’t perturb a given parameter, then the causality graph will not find the related connection for the next iteration, even if such connection should emerge due to change in the policy.

My other concern is regarding limited evaluations. In Algorithm 1 line 16, the number rollouts collected from the simulation (roughly) matches the number of simulation parameters. One could estimate the gradient via finite difference and use line search with roughly the same amount of compute. Perhaps this approach will break for more complex domains, but it seems a natural thing to try in this case. Second, the task has only four actions (starting position of the pusher, hitting angle, and velocity), which might be directly learnable (or fine-tunable) on a robot with the same data budget. Therefore, the task serves as a great test domain for illustrating the method, but does not help understand how the method scales to harder problems. For example a more complex sim-to-sim example could help understand scalability of the method better. Third, the paper uses the same hyperparameters ($MaxIter$, $N$, $M$) for all baselines, and no justification is given for the particular choice. Instead, it would be better to tune the baselines separately and report performance as a function of environment rollouts (both sim and real separately).

Minor comment:
* I think there should not be equality on line 135 between $d_\tau$ and $f_\phi$. This is what we optimize for, but is never exactly true.
* Typos in Algorithm 1 and 2: casualty → causality.


**Quality Of The Limitations Section:**

Additional details required

**Questions For Rebuttal:**

* Please elaborate on the importance/practicality of Casuality-Guided Domain Randomization. Perturbing parameters that don’t change rollouts does not hurt, and if we leave out such perturbations, we cannot learn potential causal relations on the next iteration. How is the perturbation magnitude $\delta_r$ for each parameter chosen?
* Evaluations are too limited. Please justify the choice of the hyperparameters for each algorithm and, in case a different sets of hypers work (clearly) better for different algorithms, report the results in terms of the number of rollouts needed on the real robot.
* If possible, please consider a more challenging domain with higher dimensional action and parameter space. Alternatively, please include a naive baseline that uses finite-difference gradients for tuning the simulation parameters, and discuss (or show) whether the policy could be optimized or fine-tuned directly on hardware with the same data budget.

**Robotics Focus:**

Sufficient demonstration on hardware

**Summary Of Paper:**

The paper proposes a method to learn causal relationships between simulation parameters and states. The output of the method is a differentiable causal graph that can be used in various ways: First, it provides insights to the system by indicating the importance of different parameters to the system’s behavior. Second, the graph can be used to devise targeted domain randomization that narrows down the number parameters to randomize. Third, since the graph is differentiable, one can directly optimize simulation parameters with gradient based optimization by minimizing the difference between simulated and real trajectories. Forth, the graph can be used as a part of sim-to-real policy optimization by iteratively collecting data from a real robot, fitting the simulation parameters, and optimizing a new policy in simulation. The proposed algorithms are evaluated on table top air hockey both in sim-to-sim and sim-to-real setting, and compared to three prior methods.

**Summary Of Recommendation:**

While the proposed approach to learn causal graphs is sound, evaluation and comparison to prior works is limited, making it difficult to asses the benefits and potential use-cases. Better evidence to support the claims made in the paper are needed to justify acceptance.

After the rebuttal phase, I have increased my recommendation score from Weak Reject to Weak Accept. The authors have provided detailed answers to my concerns. However, I remain slightly unsure of the practical significance of the work. This may be due to my limited knowledge of related work, which is reflected in my confidence score.

---

### Author Response · Authors · 2023-08-11
**Summary of Rebuttal**

Dear reviewers:

Thank you all for the constructive feedback and encouraging review. In addition to the detailed comments to each reviewer's questions, we would like to highlight our modifications including the new experiments we added during the rebuttal phase.

* Added a sim-to-real Double-bouncing-ball experiment (Appendix D)
* Added a sim-to-sim Push-I experiment (Appendix E)
* Added 3 ablation studies for Air-hockey experiment (Appendix F)
* Added an additional literature review section (Appendix G)
* Fixed typos and refined wording in the introduction

We would appreciate you kindly checking our response. Please do not hesitate to contact us if there are other clarifications we can offer. Thanks!

Best,

Authors

---

### Author Response · Authors · 2023-08-15
**Letter to AC**

Dear AC,

After a thorough discussion with reviewer 7MJV, reviewer 7MJV has confirmed that we have addressed all of the concerns through clarifications and additional experiments. However, we haven't received any feedback from reviewer mBeP, who initially recommended a weak reject. As the discussion period ends today, would you please remind reviewer 7MJV to update the rating accordingly and reviewer mBeP to acknowledge receiving our rebuttal response and ask any further questions?

Best,

Authors.

---

### Author Response · Authors · 2023-08-16
**General Response**

We thank all reviewers for their constructive and thoughtful feedback. Reviewers agreed that **the paper is well-written and solves an important problem** (reviewer UgPf, Ltn4, mBeP, and 7MJV), **the proposed method is novel and interesting** (reviewer UgPf, Ltn4, and 7MJV), **the experiments are well-executed** (reviewer UgPf, Ltn4, and mBeP). Here we hope to restate our contributions:

1. In contrast to prior work that requires extensive domain knowledge to choose the set of environment parameters, we propose a **novel causality-guided parameter estimation framework**, **COMPASS**, to identify the most relevant environment parameter in an automated manner and align the simulator with the real world, thus improve agent performance in the real world.
2. We design a fully-differentiable model that explicitly embeds that causal structure to provide **better interpretability** and **prune the search space of parameters**. By pruning the search space, COMPASS improves the real and simulated **sample efficiency** compared with existing methods.
3. We empirically evaluate our method in three simulated and real-world environments. Empirical results show that our proposed method **consistently outperforms baselines in terms of trajectory alignment and task success rate** with the same amount of real and simulated rollouts.

To address the reviewers' questions, we have made our best efforts during the rebuttal period to add the following:
* a new sim-to-real double-bouncing-ball experiment to follow reviewer mBeP's suggestion to have more than 1 task in the real world.
* a new sim-to-sim contact-rich push-I experiment to address reviewer 7MJV's concern that the original task may be directly solvable from real data.
* three additional ablation studies:
    1. differnt rollout size $N$ and $M$: Address reviewer 7MJV's concern about the choice of hyperparameter.
    2. different initial parameters: Answer reviewer UgPf's question about the sensitivity.
    3. different sparsity weight $\lambda$: Answer reviewer Ltn4's question about the regularization term.
* a supplementary literature review section to discuss reviewer mBeP's question about our relationship to policy generalization and adaptation work in quadruped locomotion literature.

We believe these materials have complemented our rebuttal well and addressed most concerns of reviewers.

However, there is one remaining concern from reviewer mBeP that policy adaptation baselines should be included in this work and show COMPASS outperforms them, which we respectfully disagree with. We restate the three main reasons here:
1. Policy adaptation and system identification (SysID) should not be viewed as competing solutions to the sim-to-real challenge. Rather, they complement each other. The aim of COMPASS and selected baselines is to optimize the training task distribution, making it more reflective of the target testing task distribution, which is also preferable for the training of adaptive policy.
2. There is vast work on meta-learning, policy adaptation, and multi-tasking in quadruped locomotion and manipulation. Introducing one or two locomotion baselines to our manipulation-focused study might not provide the clarity or evidence the reviewer suggests, especially given the different evaluation metrics for generalization and SysID.
3. SysID approaches have a deep-rooted tradition in robotics and remain at the forefront of research. For our paper, we opted for the most fitting and directly relevant baselines, such as Tune-Net, EXI-Net, and NPDR, which have been highlighted in recent CoRL publications. Notably, many policy adaptation algorithms, including the one suggested by the reviewer (Kumar et al.), integrate a SysID component similar to our baselines.

In conclusion, we proposed a novel causality-guided parameter estimation framework and have demonstrated our paper's unique contributions through extensive experiments and ablation studies in three different tasks in the simulation and the real world. We firmly believe that the framework we proposed to bridge the simulator and the real world is of wide interest to the robot learning community. Thank all reviewers, area chair, and program chair again for the time and effort invested in reviewing this paper!

---

### Decision · Program_Chairs · 2023-08-30

**Decision:**

Accept (Poster)

**Comment:**

Thank you for your submission to CoRL 2023. The reviewers have spent considerable time working with this paper and the current version is recommended for acceptance.

Please follow up with promised improvements, experiments and additional suggestions from the discussion phase. These include in particular the extended evaluation and comparison against simpler baselines.